# The functional logic of odor information processing in the *Drosophila* antennal lobe

**Aurel A. Lazar**‡*, **Tingkai Liu**‡, **Chung-Heng Yeh**‡

Department of Electrical Engineering, Columbia University, New York, NY, United States of America

‡ The authors' names are listed in alphabetical order.
* aurel@ee.columbia.edu

**Data Availability Statement:** All data and code are open access and are available in a public repository.(https://github.com/TK-21st/AntennalLobeLLY22).

## Abstract

Recent advances in molecular transduction of odorants in the Olfactory Sensory Neurons (OSNs) of the *Drosophila* Antenna have shown that the *odorant object identity* is multiplicatively coupled with the *odorant concentration waveform*. The resulting combinatorial neural code is a confounding representation of odorant semantic information (identity) and syntactic information (concentration). To distill the functional logic of odor information processing in the Antennal Lobe (AL) a number of challenges need to be addressed including 1) how is the odorant *semantic information* decoupled from the *syntactic information* at the level of the AL, 2) how are these two information streams processed by the diverse AL Local Neurons (LNs) and 3) what is the end-to-end functional logic of the AL?

By analyzing single-channel physiology recordings at the output of the AL, we found that the Projection Neuron responses can be decomposed into a *concentration-invariant* component, and two transient components boosting the positive/negative concentration contrast that indicate onset/offset timing information of the odorant object. We hypothesized that the concentration-invariant component, in the multi-channel context, is the recovered odorant identity vector presented between onset/offset timing events.

We developed a model of LN pathways in the Antennal Lobe termed the differential Divisive Normalization Processors (DNPs), which robustly extract the *semantics* (the identity of the odorant object) and the ON/OFF semantic timing events indicating the presence/absence of an odorant object. For real-time processing with spiking PN models, we showed that the phase-space of the biological spike generator of the PN offers an intuit perspective for the representation of recovered odorant semantics and examined the dynamics induced by the odorant semantic timing events. Finally, we provided theoretical and computational evidence for the functional logic of the AL as a robust *ON-OFF odorant object identity recovery processor* across odorant identities, concentration amplitudes and waveform profiles.

## Author Summary

A major challenge in the study of the *Drosophila* early olfactory sensory system is to determine how an odorant object (the smell of a rose) is reliably identified in the face of fluctuations of the concentration amplitude of the molecules within the odorant plume. More

**Funding:** The research reported here was supported by AFOSR under grant #FA9550-16-1-0410 (AAL, CHY), DARPA under contract #HR0011-19-9-0035 (AAL, TL) and NSF under grant #2024607 (AAL). The funders had no role in study design, data collection and analysis, decision to publish, or preparation of the manuscript.

**Competing interests:** The authors have declared that no competing interests exist.

fundamentally, the question arises how semantic information, often associated with subjective perception, can be characterized. To address this challenge, we leveraged the unique combinatorial odorant code of *Drosophila* and presented a formal treatment of the identity of an odorant object as its semantic information (or semantics for short). Grounded in the physiology of the fly brain, we identified the functional roles played by Local Neurons in the fruit fly Antennal Lobe in the recovery of the semantics and the onset/offset timing information (or semantic *timing*). Our model of the Antennal Lobe circuit is built with a highly versatile canonical model of neural computation—the differential Divisive Normalization Processor.

## 1 Introduction

The early olfactory system of the fruit fly senses a complex odorant landscape [1] with a diverse *odorant object identity* and and an ever changing *odorant concentration waveforms* [2]. Due to the multiplicative coupling of identity and concentration in the olfaction sensory periphery [2, 3], the combinatorial neural code of the Olfactory Sensory Neurons (OSNs) in the fruit fly (as well as in other invertebrates [4] and vertebrates [5]) confounds the two information streams. To support rapid and robust odorant recognition, associative learning, and other cognitive tasks [6–8] in higher brain centers, odor information processing in the Antennal Lobe (AL) recovers odorant object identity (encoded by the Projection Neurons (PNs), the output neurons of the AL) unambiguously in the face of concentration fluctuations.

Unlike in other sensory systems, the identity information of (odorant) stimuli is directly characterized by the set of odorant receptors as a combinatorial code [9]. This code has been systematically extracted and evaluated from steady-state OSN recordings [10]. To study both single glomerulus and multi-glomeruli AL responses, we followed [2], and defined the odorant object stimulus as

$$([\mathbf{b}]_{ron} \cdot u(t), \ [\mathbf{d}]_{ron}),$$

where the odorant concentration waveform $u(t)$ is multiplicatively coupled with the identity of the odorant object represented by the binding $[\mathbf{b}]_{ron}$ and dissociation rates $[\mathbf{d}]_{ron}$ of a given odorant $o$ interacting with the $n$-th OSN expressing the receptor type $r$, $r = 1, \ldots, R$, (see Fig 1A).

Critically, for a given set of receptors, assuming that all OSNs expressing the same receptor type have the same binding and dissociation rates (hence dropping the index $n$), the identity of the odorant object can be represented as the 2-tuple of vectors $\mathbf{b}_o = [[\mathbf{b}]_{1o}, \ldots, [\mathbf{b}]_{Ro}]^T$ and $\mathbf{d}_o = [[\mathbf{d}]_{1o}, \ldots, [\mathbf{d}]_{Ro}]^T$. In another word, the *semantic* information (identity), of the odorant object at the level of odorant receptors is entirely captured by the binding and dissociation rates (vectors) above. Additionally, the odorant semantic information is independent of the *syntactic* information embedded in the odorant concentration waveform. The confounding of the odorant *semantic* and *syntactic* information is a key feature of the odorant object stimuli, that is hypothesized here to be, disentangled by the AL circuit.

The recovery of odorant object identity at the Antennal Lobe level calls for decoupling the semantic information from the syntactic information carried by the amplitude of the concentration waveform. The question arises here whether the AL circuit can generate concentration-invariant multi-channel PN responses both between random encounters with a given odorant at different concentration levels, as well as at temporally fluctuating concentration levels [11–13]. However, to the best of our knowledge, no unified theoretical framework that accounts

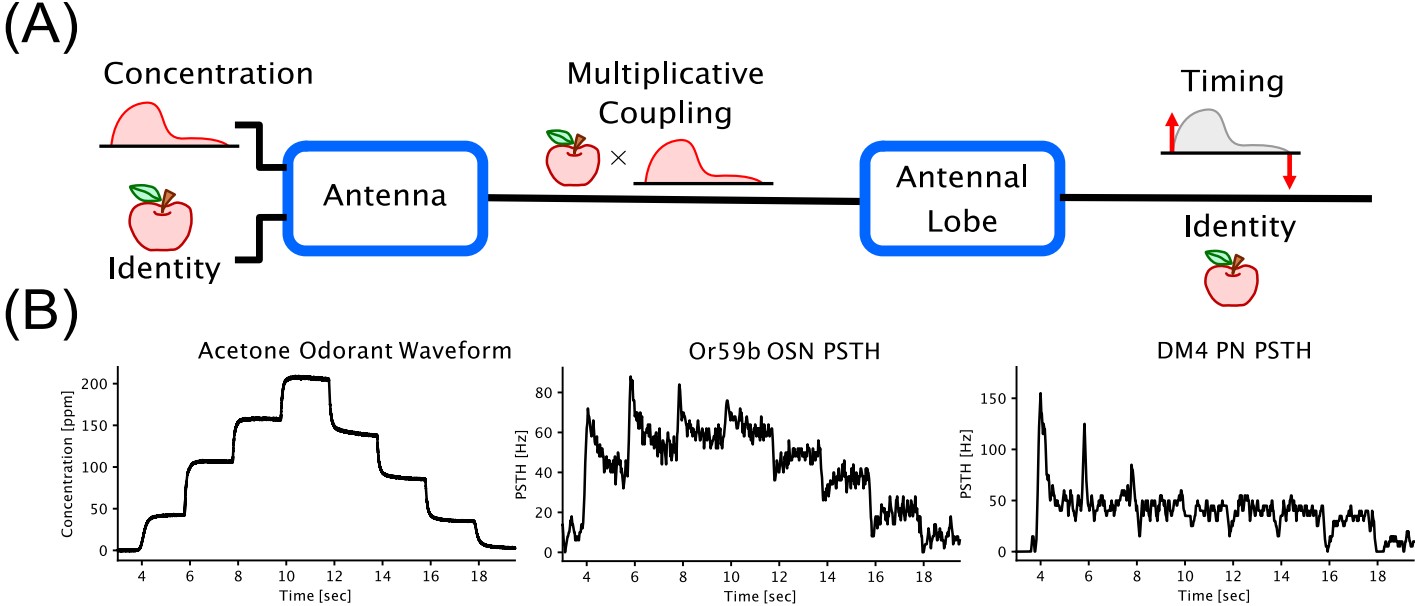

**Fig 1. Encoding and odor signal processing in the early olfactory system of the fruit fly brain.** (A) Computational logic of the olfactory system. Encoding of odorant waveforms in the Antenna is confounded between odorant object identity and odorant concentration waveform. For robust downstream odor signal processing, recognition and associative learning, the Antennal Lobe transforms the confounded Antenna odorant representation into the identity and the on/off timing information of the odorant object. (B) Input/Output (I/O) characteristics of the early olfactory system along a single channel. Shown from left to right are Acetone staircase odorant concentration waveforms, Or59b OSN Peri-Stimulus Time Histograms (PSTHs) and DM4 PN PSTHs, respectively. Data taken from [35].

for concentration invariance across receptor channels, across time and across odorant object encounters have been proposed [14–18]. Beyond accounting for concentration-invariance, it remains unsettled how PN responses (in particular uni-glomeruluar PNs) across multi-channels characterize the odorant object identity [19–22], and what role concentration information plays in multi-channel odor information processing.

From a structural stand point, the axons of OSNs expressing the same receptor-type [3, 23] and the dendrites of downstream PNs form ∼ 50 glomeruli [23–25], i.e., a set of parallel *channels* of odor information processing identified by their receptor type [6]. An example input/output map along a single DM4 glomerulus (channel) is shown in Fig 1B. Importantly, computations both within and between channels in the Antennal Lobe are facilitated by a diverse set of Local Neurons (LNs) [14, 26–30] that are differentiated by their innervation patterns (within vs. across glomeruli), neurotransmitter types, excitatory/inhibitory synapses, and innervation targets. Distilling connectivity patterns from the connectomes of both the adult [31] and larva [32] *Drosophila* and from previously known neurotransmitter profiles of the LNs [33], we considered 3 main LN cell-types: 1) pre-synaptic pan-glomerular (innervating all glomeruli) inhibitory LNs (Pre-LNs), 2) post-synaptic uni-glomerular (innervating a single glomerulus) excitatory LNs (Post-eLNs), and 3) post-synaptic uni-glomerular inhibitory LNs (Post-iLNs). Note that LNs are connecting pre-/post-synaptically to the OSN-to-PN synapses in each glomerulus. While additional LN types both in terms of innervation patterns and neurotransmitter profile exist [12, 31, 32, 34], we found that these three LN types, and the uni-glomerular PNs (henceforth referred to simply as PNs) alone, are sufficient for disentangling the syntactic and semantic information streams of the odorant stimuli. The functional significance of other LN/PN types remains an open question.

Using electrophysiology recordings of the input/output neurons [35] of the DM4 glomerulus for staircase Acetone odorant concentration stimuli, we found that the OSN and PN responses

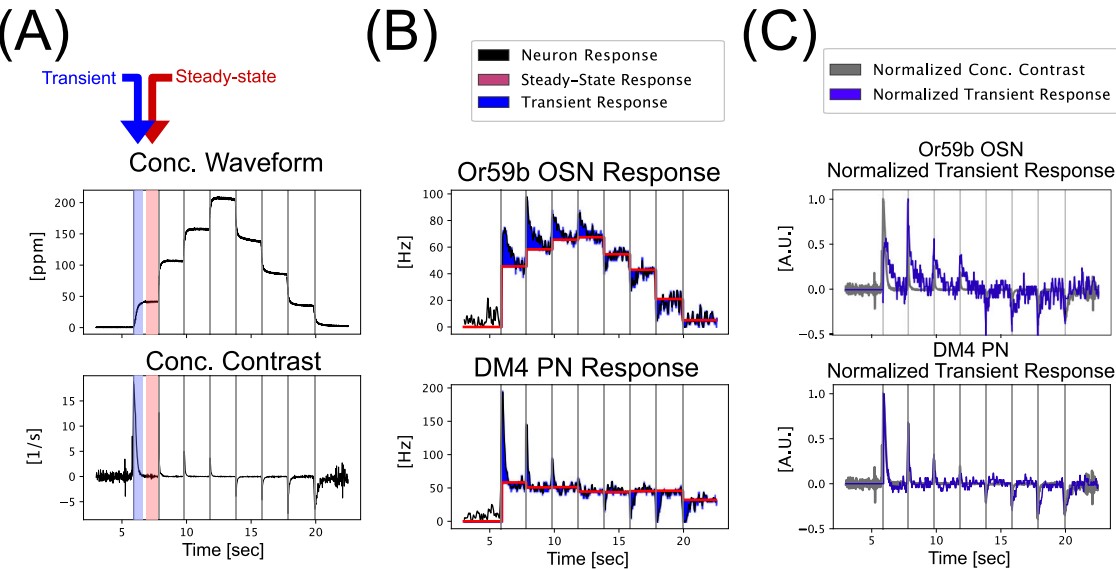

**Fig 2. Essential temporal encoding/processing characteristics of the DM4 PNs.** (A) Acetone concentration waveform (first published in [35]) and concentration contrast. (B) Or59b OSN and DM4 PN spike train data in response to an Acetone staircase waveform. Each neuron response is decomposed into steady-state and transient responses—where the steady-state response is computed as the average PSTH within a 500 miliseconds window before a jump in odorant concentration. The transient response is the residual obtained by subtracting the steady-state response from the overall response. See the text for more details. (C) Or59b OSN and DM4 PN normalized transient response compared against normalized concentration contrast. Note that the PN transient response agrees very well with the concentration contrast.

can be decomposed into linear superpositions of a piecewise constant component and a transient component (Fig 2B). For the single-channel PN response shown in Fig 2B(bottom), the piecewise constant response component exhibits a significantly reduced dependency on the amplitude of the concentration waveform (see also **S1B Fig in** S1 Appendix(bottom)). Furthermore, the amplitudes of the piecewise constant components show a much lower degree of variability for the PN response as compared to the OSN response (see **S5 Fig in** S1 Appendix). Henceforth, we refer to this response feature of reduced concentration dependency as *concentration-invariance* [4, 5, 11–13]. Furthermore, the transient PN responses show a much higher correlation with the *concentration contrast* (Fig 2C and **S1B Fig in** S1 Appendix(middle), a response feature henceforth referred to as *contrast-boosting*. (See Section 4.2 in Methods for the definition of concentration contrast.) We note that, the temporal integral of the concentration contrast amounts to the value of the absolute concentration amplitude in log-space (see also Eq (7) in Section 4.2). Importantly, as PNs are spiking neurons, the concentration-invariant response component corresponds to oscillations around stable attractors in the phase-space (see **S1 Table in** S1 Appendix for a brief description.) of the PN Biophysical Spike Generator (BSG) model. Similarly, the transient response components correspond to temporary perturbations of the limit cycles of PN BSG in response to ON/OFF event timing information extracted from the concentration waveform of an odorant with known identity (Acetone). As such, the separation of concentration-invariance and contrast-boosting of PN response can be seen directly in the phase-space of the PN BSG models (See Section 2.2 for more details).

Drawing inspiration from the 2D encoding [2] model of the OSN transduction process, we introduced a new class of biophysical models termed the differential Divisive Normalization Processors (DNPs) that are functionally similar to the (convolutional) Divisive Normalization Processors [36, 37] previously proposed to describe the fly early visual system. To characterize both single-channel and multi-channel Antennal Lobe circuits, we advanced temporal and

spatio-temporal variants of differential DNPs, respectively. For single-channel AL circuits, by modeling the Pre-LN, Post-iLN and Post-eLN pathways as 3 parallel temporal differential DNPs, we found that Pre-LN inhibition enhances the *concentration-invariance* of the PN responses. Furthermore, Post-eLN excitation and Post-iLN inhibition capture odorant concentration waveform onset and offset timing information, strongly boosting the contrast of the PN transient responses from the ones observed in the OSN responses.

By scaling from the single-channel to the multi-channel AL circuit, we showed that the concentration-invariance and contrast boosting PN response characteristics can be similarly achieved by differential DNPs across all channels. From a functional perspective, we hypothesized that the concentration-invariant component of the PN population responses are, in fact, a recovery of the odorant semantics as represented by the odorant *affinity rate* vector $\mathbf{b}_o \oslash \mathbf{d}_o$ (where $\oslash$ indicates element-wise division) [2]. We demonstrated that spatio-temporal differential DNP along the Pre-LN pathway is critical for the recovery of the odorant semantics at the level of PNs. Thus, *semantic* information of odorant object identity, represented as a set of concentration-invariant stable attractors in the PN BSGs phase-space, is readily available across glomeruli for downstream processing in the Mushroom Body and Lateral Horn. Furthermore, the contrast-boosting component of PN responses signals ON/OFF event timing of changing odorant semantics (i.e., identity of the odorant object) that is critical for real-time odor signal processing. Together, the multi-channel AL circuits disentangle the confounded semantic and syntactic information streams at the output of the Antenna, and encode both the identity and the timing information of the odorant objects into PN output spike trains.

Finally, we showed that the computation of ON/OFF timing and the recovery of the identity by the AL circuit are robust across odorant identity, concentration and waveform profile. Thus, the functional logic of the AL is that of an *ON-OFF odorant object identity recovery* processor, enabling rapid and robust odorant recognition and olfactory associative learning downstream.

## 2 Results

In Section 2.1, we investigated odor information processing in single-channel and multi-channel Antennal Lobe circuits. For the single-channel circuit of the AL (see Section 2.1.1), three LN pathways (presynaptic inhibition, postsynaptic inhibition & excitation) were modeled as *temporal differential* DNPs. We showed that the 3 differential DNPs achieve concentration-invariance and contrast-boosting in parallel, which closely reproduce the characteristics of the temporal response observed in PN electrophysiological recordings. For the multi-channel AL (see Section 2.1.2), we extended the temporal differential DNP describing the Pre-LN pathway to a *spatio-temporal differential* DNP that describes the lateral inhibition mechanism of the panglomerular Pre-LNs [3, 15]. We showed that the multi-channel model is also able to capture the concentration-invariance and contrast-boosting response features. The multi-channel circuit encodes ON/OFF timing-events that capture onset/offset changes of odorant concentration waveforms and are synchronized across Post-eLN/Post-iLN pathways (channels). Additionally, the multi-channel circuit produces concentration-invariant responses across channels and time.

In Section 2.2, we argued that the odorant *semantic information* (odorant object identity) represented by the affinity vector, can be recovered by processing the PN response. In the phase-space of PN Biophysical Spike Generators (BSGs), the semantic information is mapped into stable attractors [38]. Similarly, we showed that the onset/offset changes captured by PN transient responses correspond to ON/OFF event timing information characterizing the odorant object, and are mapped into temporary limit cycle perturbations around stable attractors.

We argue that the functional logic of the AL is that of *an ON-OFF odorant object identity recovery processor*.

Finally, in Section 2.3, we showed that the functional logic of the AL circuit is robust with respect to different odorant identities, concentration amplitudes and concentration waveform profiles.

## 2.1 Concentration-invariance and contrast-boosting in the antennal lobe

**2.1.1 Concentration-invariance and contrast-boosting in a single-channel antennal lobe circuit.** Recall that, based on physiological recordings, analysis of the temporal response of PNs innervating the DM4 glomerulus in Fig 2 reveals two key features of odor signal processing: *concentration invariance* and *contrast boosting*. Therefore, to understand the odor information processing in the Antennal Lobe circuits, we sought to systematically determine the DM4 PN temporal response characteristics put into effect by Pre-LNs, Post-eLNs and Post-iLNs.

Since the physiological recordings of uniglomerular PN responses only pertain to a single receptor type (here Or59b and, correspondingly, the DM4 glomerulus), we began by developing a model of single-glomerulus AL circuit. From the modeling perspective, the Calcium Feedback Loop of the Odorant Transduction Process (OTP) model of the OSN produces a transient response similar to that observed in the PN physiology data [2]. As such, the calcium feedback in OTP model serves as a starting point for circuit component models of the Antennal Lobe.

The OTP model can be described more generally by *differential* DNPs, where for a given ratio of the ligand-bound receptors $v(t)$, the gating variable of the co-receptor channel $x_1(t)$ can be described by the equation [2]:

$$\frac{dx_1}{dt} = \mathcal{T}_1[v] \cdot (1 - x_1) - \beta_1 \cdot x_1 - \mathcal{T}_2[x_1] \cdot x_1, \tag{1}$$

where functionally $\mathcal{T}_1[v]$ describes an amplification of $v(t)$, and $\mathcal{T}_2[x_1]$ describes the strength of a state dependent feedback loop. The general model in Eq (1) describes the transition rate from state $1 - x_1$ to state $x_1$ as $\mathcal{T}_1[v]$, and the transition rate from $x_1$ to $1 - x_1$ as $\beta_1 + \mathcal{T}_2[x_1]$.

We note that the critical point (steady-state) solution of Eq (1) (assuming that it exists)

$$x_1(\infty) = \frac{\mathcal{T}_1[v]}{\beta_1 + \mathcal{T}_1[v] + \mathcal{T}_2[x_1]}, \tag{2}$$

functionally reminds us of the Divisive Normalization Processors [36, 37] previously proposed to describe contrast gain control in the fly early visual system. Since the OTP model [2] was cast as a dynamical system, we refer to the model in Eq (1) as the *differential* DNP. Note that, the *feedback* differential DNP model in Eq (1) can be readily modified to a *feedforward* configuration (by replacing the feedback signal $\mathcal{T}_2[x_1]$ with an operator on the input signal $\mathcal{T}_4[v]$) satisfying the differential equation:

$$\frac{dx_2}{dt} = \mathcal{T}_3[v] \cdot (1 - x_2) - \beta_2 \cdot x_2 - \mathcal{T}_4[v] \cdot x_2. \tag{3}$$

Based on the connectivity structure of the differential DNP, we modeled the Pre-LN inhibition of the OSN Axon-Terminal (see Fig 3A1). Here, the Pre-LN spike train inhibits the vesicle release in the OSN Axon-Terminal in a feedforward configuration. The OSN-to-Post-LN (here Post-LN refers to Post-eLN or Post-iLN) synapse models (see Fig 3A1) expand upon the calcium feedback loop component of the OTP model, and introduce additional dynamics into

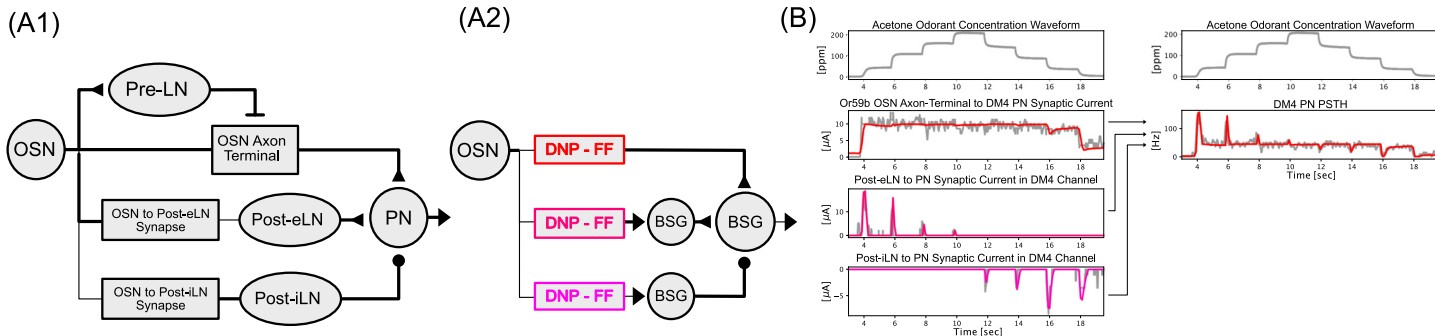

**Fig 3. An Example of Concentration-Invariance and Contrast-Boosting in a Single-Channel Antennal Lobe Circuit.** (A1) Model architecture of a single-channel (glomerulus) Antennal Lobe circuit. (A2) Functional equivalent model of the single-channel AL circuit in (A1) with each branch replaced in part by a differential DNP. For Post-LN pathways, BSGs are included to capture the spike generation of Post-LNs. (B, left) Input and output relationships for each of the three branches shown in (A1, A2). The odorant concentration waveform and Pre-LN, Post-eLN and Post-iLN to PN synaptic currents estimated from physiology recording data are shown in grey. Response of each DNP (and DNP-BSG cascade) pathway in (A2) is shown in red. (B, top right) Acetone staircase concentration waveform. (B, bottom right) Model DM4 PN PSTH (red) compared with the DM4 PN physiology recording (grey).

the input processor $\mathcal{T}_3[v]$ in Eq (3). Based on the choice of parameter values (see Section 4.1 in the Methods section for details), the feedforward model of the OSN-to-PostLN synapses provides the flexibility to strongly respond to either input onset or offset. Finally, all other synapses in the single-channel AL circuit (e.g., OSN Axon-Terminal to Pre-LN, OSN Axon-Terminal to PN, Post-eLN to PN and Post-iLN to PN are modeled as ionotropic [39], and all biophysical spike generators (BSGs) are modeled as noisy Connor-Stevens point neurons (see [2] and Section 4.1 in Methods for more details). For modeling details of the single-channel AL circuit, refer to **Single-Channel Antennal Lobe Circuit** in Section 4.1.

Since the Local Neuron pathways (OSN Axon-Terminal to Pre-LN synapse and Pre-LN to OSN Axon-Terminal vesicle inhibition, OSN to Post-eLN synapse, OSN to Post-iLN synapse) in Fig 3A1 can each be modeled as differential DNPs, the single-channel (glomerulus) AL circuit reduces to 3 subcircuits each modeled by a differential DNP as shown in Fig 3A2. Note that as opposed to Pre-LN inhibition of the OSN Axon-Terminal which, in turn, injects synaptic currents directly into the downstream PNs, the OSN-to-Post-LN synapses connect to PNs via BSGs that describe the spike generation process of the Post-LNs. As mentioned above, all BSGs, are modeled here as Connor-Stevens point neurons.

Following the procedure described in Section 4.2, we first estimated the input synaptic current to the DM4 PN based on physiology recordings. The synaptic current is then linearly decomposed into concentration-invariant piecewise constant and ON/OFF contrast boosting transient signal components as shown in Fig 3B(grey). We then optimized the parameters of the three differential DNPs in Fig 3A2 by independently minimizing the angular distances between the output of each differential DNP (corresponding to synaptic current from each LN pathway into the PN as shown in Fig 3B(left, red)) and the corresponding ground truth signal components (Fig 3B(left, grey)) (for more details, see Section 4.3). Due to the high number of parameters of the AL circuit, we developed a 2-step optimization procedure (see Section 4.3 in Methods) that combines an initial random sampling of the parameter space and a fine-tuning global optimization step. The resultant temporal response of each LN pathway is shown in Fig 3B(left), and the overall DM4 PN response is shown in Fig 3B(bottom right). We observed that the DM4 PN response is well captured by the model, and that the differential DNP model is capable of capturing concentration-invariance and contrast-boosting along the Pre-LN and the Post-eLN/Post-iLN pathways.

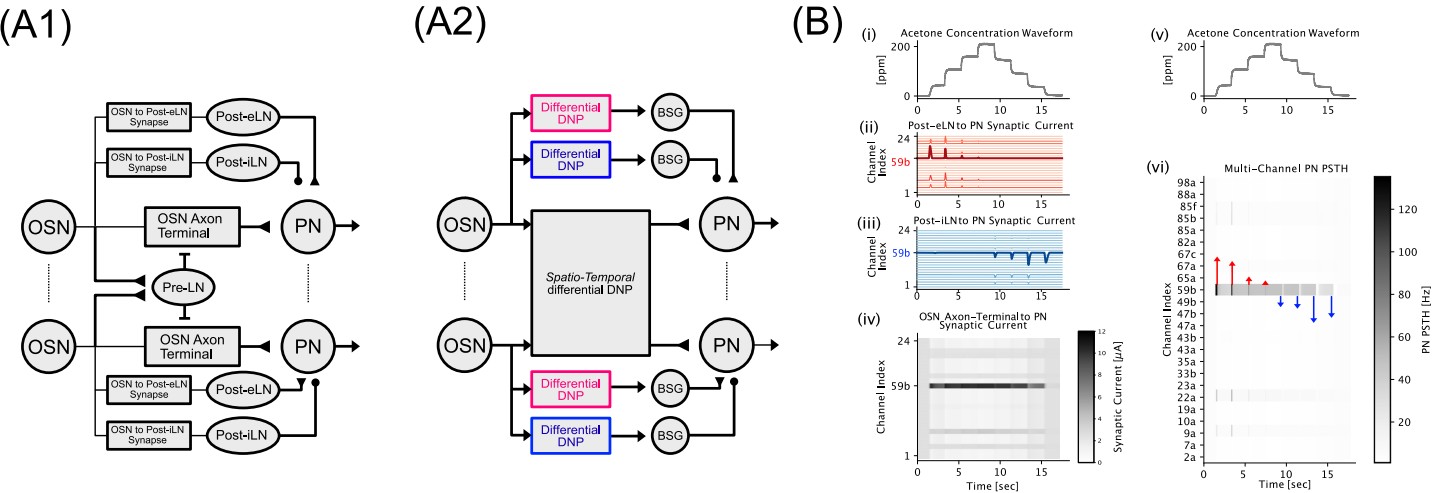

**Fig 4. An Example of Concentration-Invariance and Contrast-Boosting in a Multi-Channel Antennal Lobe Circuit.** (A1) Model architecture of multi-channel AL circuit. (A2) Functionally equivalent model of the multi-channel AL circuit, with each branch in (A1) replaced in part by a corresponding differential DNP model. (B) Input and output relationships of the multi-channel AL circuit. (B[i,v]) Acetone staircase odorant concentration waveform. (B[ii]) Post-eLN to PN synaptic currents. Different hues of red indicate strengths of synaptic current, with strongest due to the Post-eLN receiving input from the Or59b OSN indicated in dark red. (B[iii]) Post-iLN to PN synaptic currents. Different hues of blue indicate strengths of synaptic current, with strongest due to the Post-iLN receiving input from the Or59b OSN indicated in dark blue. (B[iv]) OSN Axon-Terminal to PN synaptic current shown as a heatmap. Different hues of grey indicate strengths of synaptic current. Refer to colorbar for scale. (B[vi]) Multi-channel PN PSTH, with synaptic current of ON and OFF pathways along Or59b/DM4 channel shown as blue and red arrows, respectively.

### 2.1.2 Concentration-invariance and contrast-boosting in a multi-channel antennal lobe circuit.

While the single-channel AL circuit discussed above showed that Pre-LNs and Post-LNs support *concentration-invariance* and *contrast-boosting*, they did not fully capture the multi-channel nature of odorant information processing in the AL circuit and the odorant semantics. From a structural perspective, recall that certain LN types, e.g., the Pre-LN, provide reciprocal inhibition to more than one glomerulus. In the current work, we assume that Post-LNs connect locally to each glomerulus, while the Pre-LN connect globally to all glomeruli. Based on these assumptions, we generalized the single-channel AL circuit described in the previous section to a multi-channel AL circuit (Fig 4A1). We extend the single-channel differential DNP model in Eq (1) to a *Global* Differential DNP model that describes multi-channel Pre-LN inhibition of OSN Axon-Terminals:

$$\frac{d}{dt}x_{3r}(t) = \alpha_3 \cdot \mathcal{T}_5[v_r] \cdot (1 - x_{3r}) - \beta_3 \cdot x_{3r} - \kappa_3 \cdot \mathcal{T}_6[\mathbf{x}_3] \cdot x_{3r}, \qquad \text{Global Feedback}$$

$$\frac{d}{dt}x_{4r}(t) = \alpha_4 \cdot \mathcal{T}_7[v_r] \cdot (1 - x_{4r}) - \beta_4 \cdot x_{4r} - \kappa_4 \cdot \mathcal{T}_8[\mathbf{v}] \cdot x_{4r} \qquad \text{Global Feedforward}$$

(4)

where $\mathbf{v}(t) = [v_1(t), \ldots, v_R(t)]^T$ represents the multi-dimensional input across channels, and, $\mathbf{x}_3(t) = [x_{31}(t), \ldots, x_{3R}(t)]^T$ and $\mathbf{x}_4(t) = [x_{41}(t), \ldots, x_{4R}(t)]^T$ represent the multi-dimensional output signals of Feedback (Feedforward) DNP across channels. For each $r = \{1, \ldots, R\}$ (the index of multi-channels of the AL circuit), $v_r(t)$ represents input of the $r$-th channel (i.e., spike train of $r$-th OSN BSG), $x_{3r}(t)$ (resp. $x_{4r}(t)$) represents output signal of Feedback (resp. Feedforward) differential DNP of the $r$-th channel (i.e., the normalized neurotransmitter concentration of $r$-th OSN Axon-Terminal).

Additionally, we assumed that models describing the same circuit components across channels were restricted to share the same parameter values. For instance, all OSN to Post-eLN synapses share parameter values across channels. As such, the total degrees of freedom for the

multi-channel AL circuits Fig 4A1 is the same as that of its single-channel counterpart Fig 3A1. For global Pre-LN inhibition, all OSN-to-Pre-LN synapses share the same parameter values, and the dendritic process of the Pre-LNs was modeled as the linear sum of synaptic currents across all synapses (see Section 4.1 for details).

The assumption of parameter sharing between circuit components in different channels enabled us to optimize the multi-channel AL circuit with the same objective function as for the single-channel circuit. Following the same optimization procedure, we optimized the parameters of the multi-channel AL circuit. The resulting input/output characteristics, shown in Fig 4B, demonstrate that the *concentration-invariance* and *contrast-boosting* response features of the single-channel model are preserved in the multi-channel model. For example, in Fig 4B[ii] and 4B[iii], the Post-eLN and Post-iLN pathways respond proportionally to the ON/OFF event timing in a time-synchronized manner across channels. On the other hand, the OSN Axon-Terminal responses are largely invariant to concentration both across time for single channels, and across channels with relative strength of synaptic current depending on the receptor type.

In conclusion, we have shown that single-channel AL circuits can be described by three temporal differential DNPs modeling Pre-LN, Post-eLN and Post-iLN pathways—each capturing the concentration-invariance and ON/OFF contrast-boosting response characteristics of the DM4 PN physiology recordings. By extending the temporal differential DNP to a spatio-temporal differential DNP, we in effect modeled the multi-channel AL circuit. We have shown that the multi-channel AL circuit computes ON/OFF event timing via the Post-eLN and Post-iLN pathways in a time-synchronized manner across channels. Additionally, the concentration-invariance is observed both across time along a single-channel, and across channels. Note that the presentation in this section is centered on staircase concentration waveforms for given odorant semantics. The processing of arbitrary odorant concentration waveforms in the multi-channel AL model is discussed in detail in Section 2.3.

## 2.2 Recovery of semantic information by the multi-channel antennal lobe circuit across the space of odorants

The odor information processing discussed in Section 2.1 focused on the single- and multi-channel response characteristics of the Antennal Lobe to arbitrary concentration waveforms of an odorant object with a known identity, e.g., Acetone. In another word, the discussion in Section 2.1 focused on capturing the response characteristics of PNs to changing *syntactic* information at the input of the early olfactory system. To fully understand the functional logic of the Antennal Lobe, we compared how different odorant identities (semantics) are represented by the PN output spike trains. For simplicity we assumed here that different odorants exhibit the same concentration waveforms. Put differently, we addressed the question how the PN responses to the transient syntactic information of the odorant concentration waveforms relate to, or can be distinguished from, the *semantic* content of the odorant object identity. Since the OSN encoding of an odorant is a confounding representation of the syntactics and semantics via multiplicative coupling of the concentration waveform with the object identity, we hypothesized that the PN response is in fact *a recovery of the identity of the odorant object*.

The amount of information characterizing the odorant semantics (object identity) contained in the response of the multi-channel AL circuit can be quantified by the angular distance between the PN response and the odorant affinity vector (see Section 4.3 for more details). The angular distance metric is minimized when the PN response and the odorant affinity are, up to a scaling factor, identical. From an optimization standpoint, as the concentration-invariant component of the PN response is due to the processing in the Pre-LN

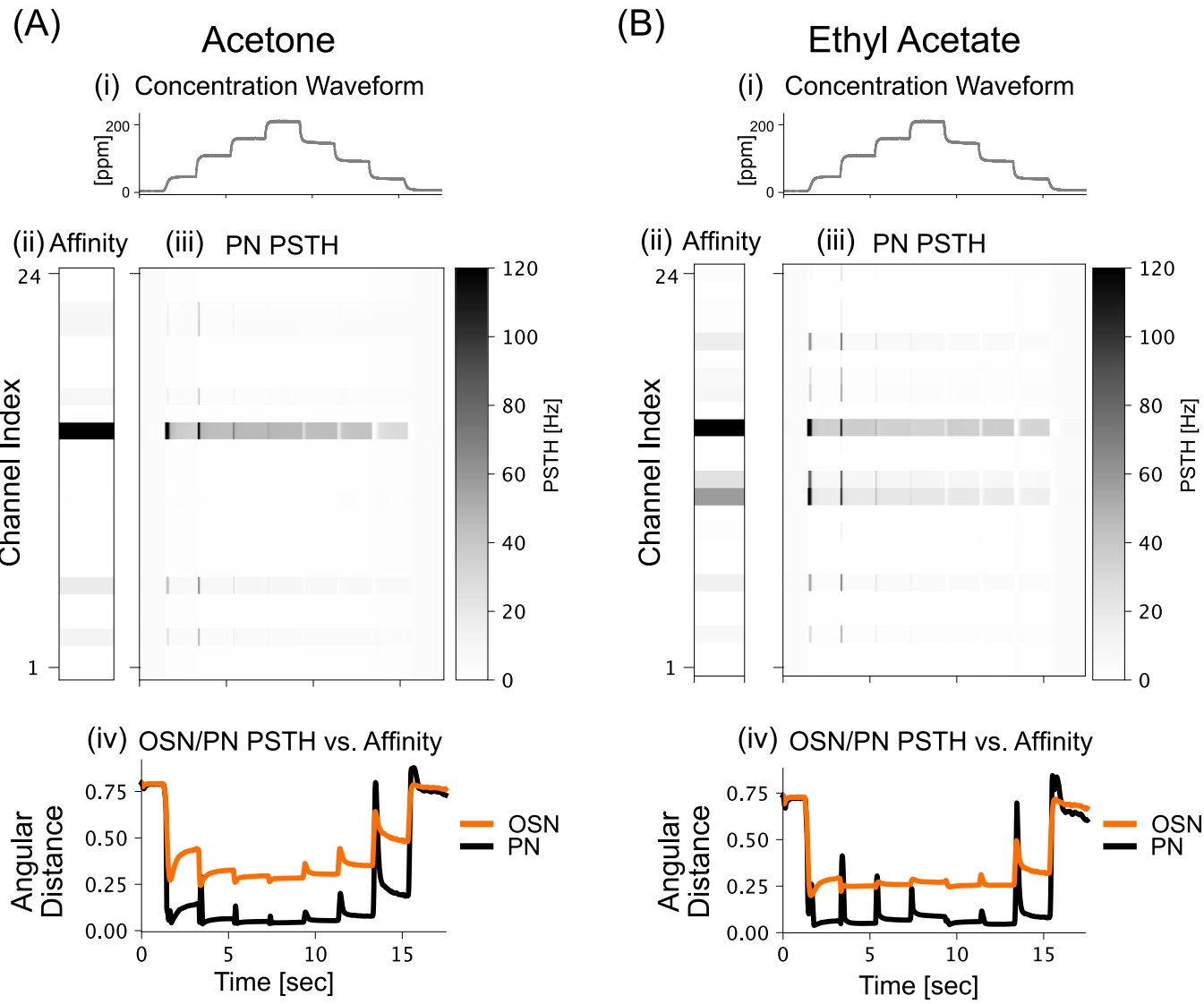

**Fig 5. Recovery of odorant semantics via angular distance minimization.** (A[i]) Acetone concentration waveform. (A[ii]) Acetone affinity vector shown as a heatmap. (A[iii]) PN multi-channel PSTH in response to Acetone input. (A[iv]) Angular distance between OSN/PN multi-channel PSTH and affinity vector shown in orange/black, respectively. Note that the angular distance between the PN multi-channel PSTH and affinity vector is lower (closer to 0) than that between the OSN multi-channel PSTH and affinity vector. (B) Similar results in (A) but shown for an Ethyl Acetate concentration waveform.

pathway, we can instead minimize the angular distance between the OSN Axon-Terminal synaptic current into PN BSGs obtained from physiology data and, model response. Note that since only the synaptic input into the DM4 PN BSGs is directly measurable from the dataset [35] as shown in Fig 3, the synaptic current for other glomeruli need to be estimated following the procedure described in Section 4.3.

Following the same two-step optimization procedure as in the previous section, we obtained the multi-channel AL circuit model parameters. The resulting PN response for Acetone and Ethyl Acetate are shown in Fig 5A[iii] and 5B[iii]. From the heatmaps of the PN PSTH, we noted that the ON/OFF transients are clearly observable for both odorant stimuli, and are all time-synchronized against the ON/OFF-set transients of the concentration waveform in Fig 5A[i] and 5B[i]. Additionally, visual comparisons between the PN PSTH responses

and the affinity vectors show their high similarity. This similarity is further highlighted by the angular distance metric shown in Fig 5A[iv] and 5B[iv]. Here the angular distances between affinity vectors and multi-channel PN responses at each time instance is close to 0 (i.e., perfect recovery) for most of the stimulus onset. For reference, the same angular distance metric is computed using the OSN response (OSN PSTH heatmap not shown), and the result, displayed in orange in Fig 5A[iv] and 5B[iv], indicates a significantly higher angular distance between the OSN response and affinity vectors. Thus, the multi-channel PN response represents the odorant object identity more closely than the multi-channel OSN response.

Theoretical analysis of the spatio-temporal differential DNP model revealed that the ability of the multi-channel AL circuit to recover odorant identity is due to the Pre-LN inhibition across glomeruli. As in Eq (2), we noted that the critical point (steady-state) solution of the Global Differential DNP (if it exists) is given by (in feedforward configuration):

$$x_{4r}(\infty) = \frac{\alpha_4 \cdot \mathcal{T}_7[v_r]}{\beta_4 + \alpha_4 \cdot \mathcal{T}_7[v_r] + \kappa_4 \cdot \mathcal{T}_8[\mathbf{v}]} . \tag{5}$$

We note that for the optimized AL circuit model parameters, $\kappa_4 \gg \alpha_4$, $\kappa_4 \gg \beta_4$, and therefore:

$$x_{4r}(\infty) \approx \frac{\alpha_4 \cdot \mathcal{T}_7[v_r]}{\kappa_4 \cdot \mathcal{T}_8[\mathbf{v}]} \propto \frac{\mathcal{T}_7[v_r]}{\mathcal{T}_8[\mathbf{v}]},$$

where $v_r$ corresponds to the spike train generated by the OSN BSG expressing $r$-th receptor type, and $\mathbf{v} = [v_1, \ldots, v_r, \ldots, v_R]^T$ corresponds the vector of spike trains of all OSNs. Given that the odorant transduction process involves a multiplicative coupling between the affinity vector and the odorant concentration profile

$$\mathbf{b}_o \oslash \mathbf{d}_o \cdot u,$$

that is in turn represented by the spike trains of all OSNs, the spatio-temporal divisive normalization provides, in steady state, a normalized odorant affinity vector

$$\approx \frac{\mathbf{b}_o \oslash \mathbf{d}_o \cdot u}{\sum_r [\mathbf{b}]_{ro}/[\mathbf{d}]_{ro} \cdot u} = \frac{\mathbf{b}_o \oslash \mathbf{d}_o}{\|\mathbf{b}_o \oslash \mathbf{d}_o\|_1} \tag{6}$$

without prior specification of the odorant identity or concentration profile [13]. We note here that the odorant semantic information can only be recovered by the AL circuit if multi-channel inhibition via the Global Pre-LN circuit is present. For single-channel AL circuits since the local Pre-LN receives input from, and provides inhibition to, a single glomerulus, the equality in Eq (6) becomes

$$\approx \frac{[\mathbf{b}]_{ro}/[\mathbf{d}]_{ro} \cdot u}{[\mathbf{b}]_{ro}/[\mathbf{d}]_{ro} \cdot u} = 1,$$

and thus the odorant identity information is lost.

While the optimization objective is formulated using OSN Axon-Terminal synaptic currents, real-time processing of odorant stimuli in biological systems involves PN spike trains. In the phase-space of the PN BSG, periodic spiking can be characterized by a set of limit cycles [38]. The identity of a limit cycle is determined by the strength of the injected current (see Fig 6). The odorant concentration value was chosen, for simplicity, to be 100 ppm. We envisioned that the identity (semantic information) of the odorant object is mapped into a unique set of PN limit cycles, termed *stable attractors*, around which the PN BSGs oscillate in an identity-dependent and concentration-invariant manner (see Fig 6A and 6B for Acetone). For

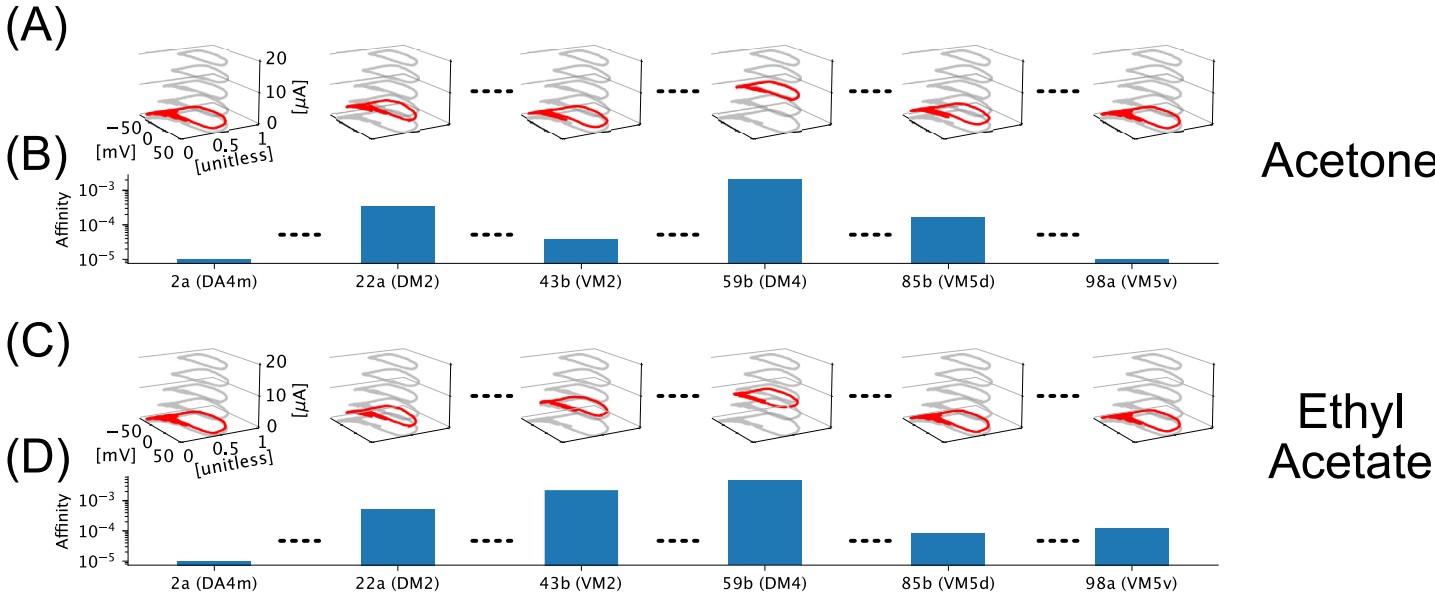

**Fig 6. Mapping of odorant semantics into PN BSG limit cycle manifolds.** (A) Limit cycles of the PN BSG for selected channels for Acetone. In each panel, the limit cycle occupied by PN in the corresponding channel is highlighted in red. Note that all 3d plots share the same axis limits and labels as the left-most plot. Other limit cycles of the Noisy ConnorStevens neuron model are shown in grey for reference. (B) Affinity vector of Acetone shown as bar plot indexed by the receptor type and the corresponding glomerulus along the x-axis. Note that the y-axis is in log-scale. (C) Limit cycles of the PN BSG for selected channels for Ethyl Acetate. Note that all 3d plots share the same axis limits and labels as the left-most plot. (D) Affinity vector of Ethyl Acetate shown as bar plot.

comparison, the stable attractors characterizing the odorant semantics of Acetone and Ethyl Acetate are shown in (Fig 6A) and (Fig 6C), respectively.

Unlike the odorant semantics, the semantic timing of the odorant objects, represented as PN transient dynamics, are to be found in the temporal traversal of limit cycle manifolds anchored in stable attractors. The term *semantic timing* refines the concept of ON/OFF timing by emphasizing the tight coupling between the *timing* and *identity* of an odorant object (see **S1 Table in** S1 Appendix for more information). In Fig 7A[ii] the DM4 PN PSTH in response to the Acetone concentration waveform in Fig 7A[i] is shown. In Fig 7A[iii] we plotted the limit cycles of the DM4 PN at four time points (labeled 1,2,3,4) corresponding to onset/stable attractor/offset/stable attractor, respectively. Results for DM4 PN response to Ethyl Acetate, VM2 PN response to Acetone and VM2 PN response to Ethyl Acetate are plotted in Fig 7B, 7C and 7D respectively. We observed that in Fig 7A, 7B and 7D, during transients, the dynamics induced by odorant timing events leads to the traversal *up (for onset) and down (for offset)* of the limit cycle manifolds. Note that, for channels that do not respond to a given odorant (e.g., VM2 PN to Acetone in Fig 7C), only the limit cycles corresponding to the resting injected current are occupied during stimulus onset.

In conclusion, we argued that the multi-channel AL circuit response exhibits the odorant semantic information provided by the affinity vector. We showed, both computationally and theoretically, that the spatio-temporal differential DNP, modeling the Pre-LN inhibition across glomeruli, supports, and is critical for (as shown in Section 4.4 in Methods) the recovery of the semantic information of the odorant across the space of all odorants. Finally, we argued that the real-time processing of odorant information is facilitated by the limit cycle manifolds of the PN BSGs. By comparing phase spaces trajectories of the multi-channel PN BSGs in response to different odorant stimuli, we showed that the odorant semantics are mapped into a set of stable attractors in the PN BSG phase-space, while event timing information extracted

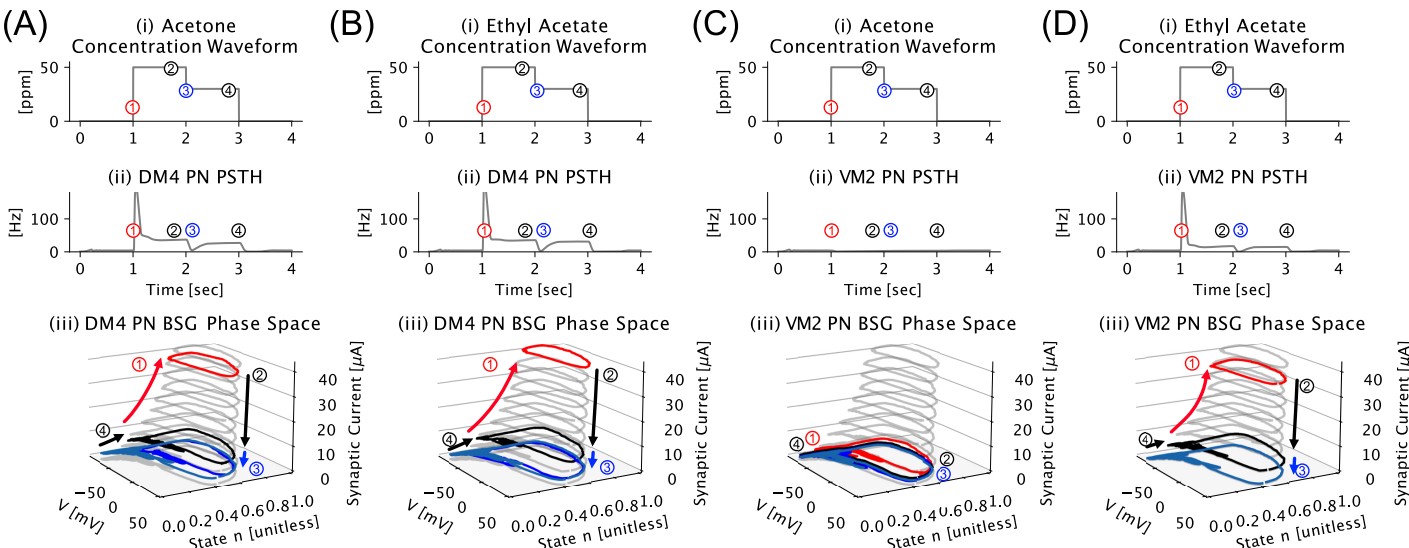

**Fig 7. Representation of semantic timing in the phase space of the PN BSG.** (A) The dynamics induced by odorant timing events in the phase space of the DM4 PN BSG. (A[i]) Acetone odorant concentration waveform with 4 segments corresponding to onset, first steady-state, first offset and second steady-state labeled as 1, 2, 3, 4. (A[ii]) DM4 PN PSTH in response to the Acetone stimulus with the corresponding PN PSTH values at time points 1, 2, 3, 4 are labeled accordingly. (A[iii]) Transitions between limit cycles of the DM4 PN BSG corresponding to the timing of the onset transient (1), steady-state (2), offset transient (3) and back to steady-state (4). (B) The dynamics induced by the Ethyl Acetate stimulus in the phase space of the DM4 PN BSG. (C-D) The dynamics induced by the Acetone and Ethyl Acetate stimuli in the phase space of the VM2 PN BSG, respectively. Note that in (C), VM2 PN does not respond to Acetone due to the low affinity of the upstream Or43B receptor to Acetone.

from the syntactics of the odorant concentration waveform leads to a temporary traversal across limit cycles to and from these stable attractors.

## 2.3 The antennal lobe is a robust ON-OFF odorant identity recovery processor

In the previous sections, we described the temporal response characteristics (concentration-invariance and contrast-boosting) and semantic information recovery of the multi-channel AL circuit. The ON/OFF event timing detection and the semantic recovery led us to hypothesize that the functional logic of the AL circuit is that of an *ON-OFF odorant object identity recovery processor*. Under such a paradigm, the Pre-LNs recover the identity of the odorant object, while the Post-eLNs and Post-iLNs extract the odorant object's ON/OFF timing information. Since the ON/OFF timing information of the odorant object is tightly connected to its identity (semantics), the timing information will be more precisely referred to as the *semantic timing* of the odorant object.

For simplicity, the discussion of concentration-invariance and contrast-boosting in Section 2.1 was limited to a fixed odorant identity, while the discussion of semantic information recovery in Section 2.2 was limited to a fixed concentration waveform across odorant identities. In this section, we demonstrated that the hypothesized functional logic of the AL circuit as an ON-OFF odorant object identity recovery processor is robust with respect to arbitrary odorant concentration waveforms and identity.

To evaluate the robustness of concentration-invariance and contrast-boosting response characteristics of the AL circuit, we evaluated the PN population response for Acetone with two different noisy concentration waveforms—the staircase in Fig 8A, and the staircase with additive white noise in Fig 8B. As shown in Fig 8A[ii], 8A[iii], 8B[ii] and 8A[iii], the detection of onset and offset timing are robust to high levels of white noise for both Post-eLN and

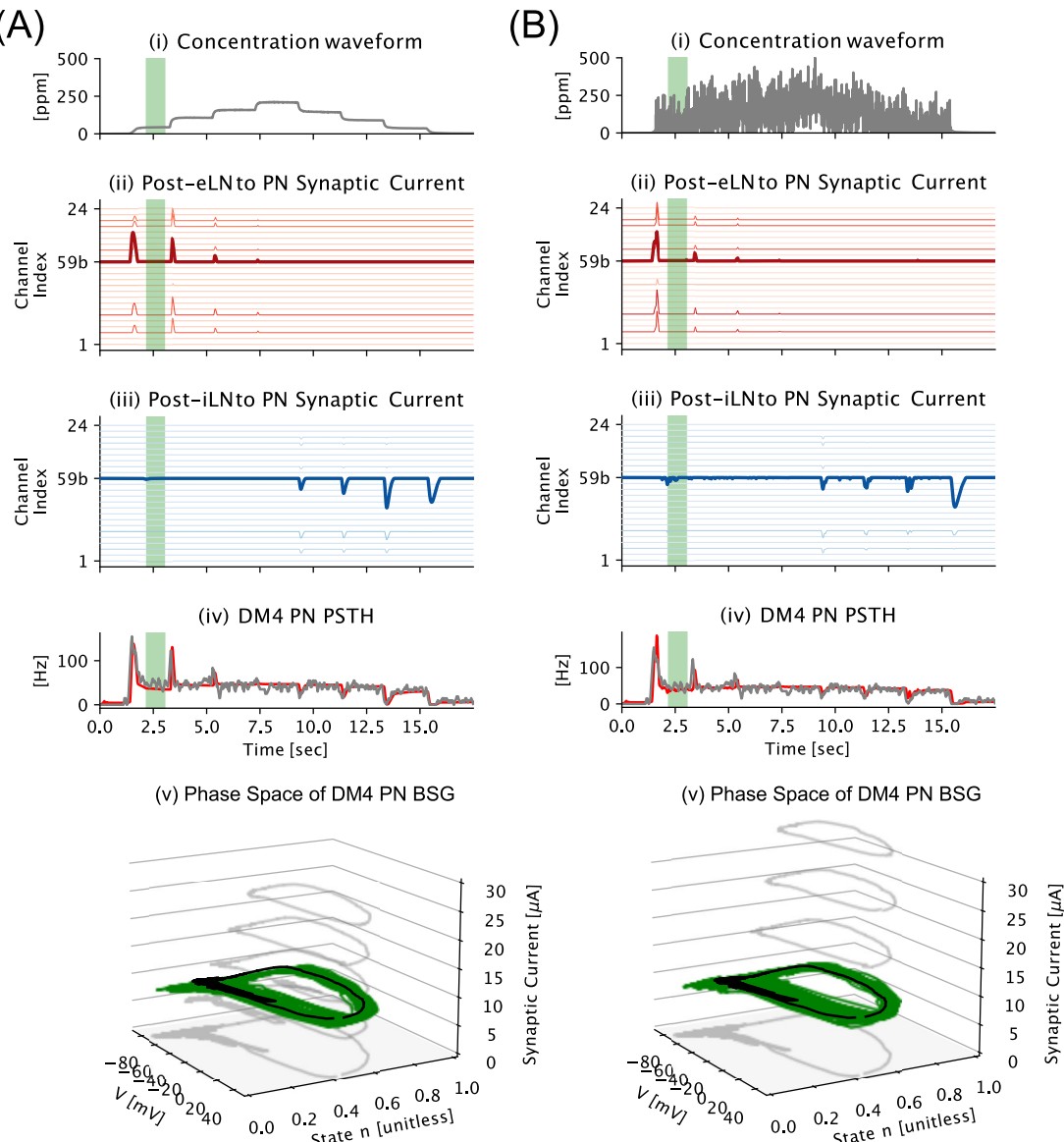

**Fig 8. Robustness of Concentration-Invariance and Contrast-Boosting of the multi-channel AL circuit.** (A [i]) Acetone staircase concentration waveform. (A [ii, iii]) Post-eLN/Post-iLN to PN synaptic current. (A [iv]) DM4 PN PSTH from physiology recording data (in grey) and model response (in red). (A [v]) Stable attractor limit cycle (in black) in DM4 PN BSG phase space and an example response trajectory of the DM4 PN BSG for 800 milliseconds between 2.2 second and 3 second, corresponding to the green shaded rectangles in (A[iv]). (B) Same as in (A) for Acetone step concentration waveform with additive white noise. Note that in the presence of strong additive white noise, the responses of the Post-eLN, Post-iLN pathways, as well as the overall DM4 PN PSTH remain similar to that in the noise-free case.

Post-iLN pathways, and similarly for more complex nonstationary signals (see **S4 Fig in** S1 Appendix). From a functional perspective, this indicates that the *semantic timing* information is reliably extracted by the AL circuit model. For the concentration-invariant component of the PN response, the fluctuations shown in Fig 8A[iv] and 8B[iv] are small, and can be viewed as perturbations of the semantic-encoding stable attractor in the phase space (see Fig 8A[v] and 8B[v]).

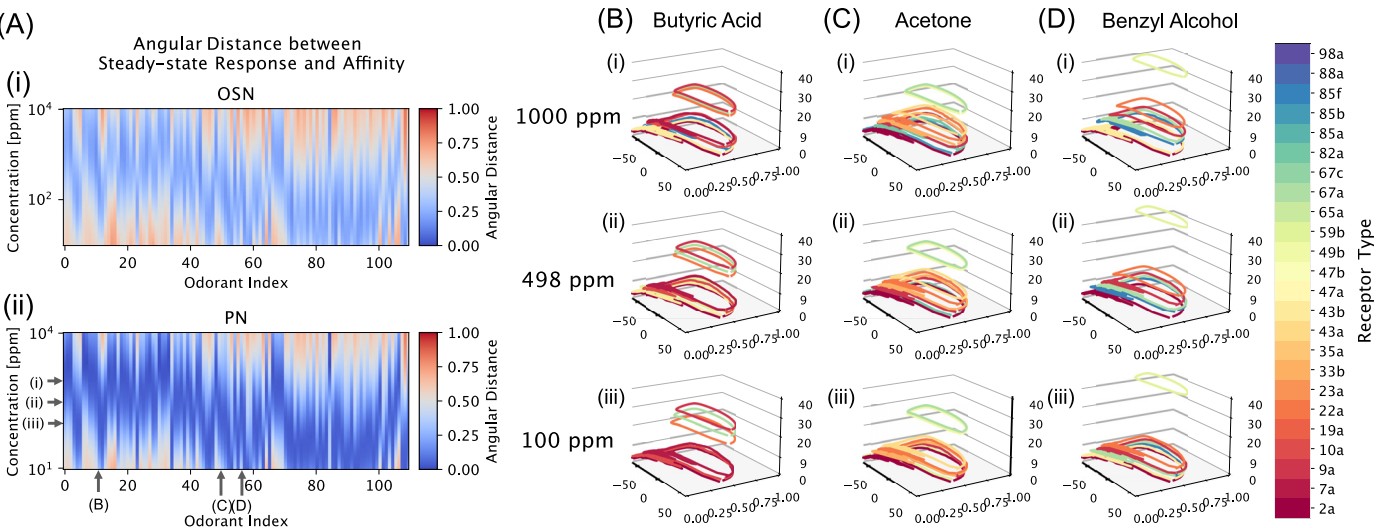

**Fig 9. Robust recovery of the identity of 110 mono-molecular odorants by the multi-channel AL circuit.** (A) Angular distance between [i] OSN and [ii] PN population responses to step concentration waveform and affinity vectors for 110 odorants for a range of $10 \sim 10,000$ [*ppm*] concentration levels (log scale). Refer to the colorbar on the right for the scale of the angular distance from low (blue) to high (red). In (A[ii]), three concentration levels and three odorants are labeled corresponding to the 9 sets of PN BSG stable attractor limit cycles shown in (B,C,D). (B) Limit cycle manifolds for all 24 channels (color-coded by channel as indicated by the legend on the far right) at for Butyric Acid concentration levels of [i] 1000 ppm, [ii] 498 ppm and [iii] 100 ppm. (C, D) Same as in (B), respectively, for Acetone and Benzyl Alcohol.

To evaluate the robustness of odorant identity recovery, we compared the PN population responses with the known affinity vectors of 110 odorants [2, 10]. The resulting angular distances between the PN population responses and affinity vectors are shown as heatmaps in Fig 9A[ii], where the x-axis represents the odorant index and the y-axis represents the concentration amplitude across 2 orders of magnitude. The color intensity is proportional to the angular distance, with blue indicating the smallest distance and red indicating the largest distance (refer to the colorbar on the right). For comparison, the angular distances between OSN population responses and affinity vectors are shown in Fig 9A[i]. Clearly, the PN population responses are closer to the values of the odorant affinity vectors. However we note that, due to the nonlinearity of the Antenna and AL circuit, for each odorant (along each column), the angular distance is highest (indicating worst recovery) at very low/high concentration levels, and is lowest at intermediate concentration levels. From a dynamics standpoint, in Fig 9B–9D, we show the semantic-encoding stable attractors for the population of 24 PNs (color-coded by channel index as shown in the color bar in Fig 9 (far right)) for 3 different odorants at 3 different concentration values. We observe that for each odorant, across concentration levels that are 1 order of magnitude apart, the stable attractors remain largely unchanged.

In conclusion, we demonstrated that the multi-channel AL circuit responses are robust with respect to arbitrary changes in the odorant concentration levels and odorant identities. Both the identity (semantics) and ON/OFF timing information (semantic timing) of the odorant object are reliably encoded by the PN spike trains. As such, we concluded that the functional logic of the multi-channel AL circuit is that of a *robust ON-OFF odorant identity recovery* processor.

## 3 Discussion

The early olfactory system of the fruit fly, while sensing a complex odorant landscape, encodes the *odorant object identity* (semantic information) and *the odorant concentration waveform*

(syntactic information) into a combinatorial neural code [2, 3]. Following [2], we argued that the identity of the odorant object is defined by the binding and dissociation rates of a given odorant-receptor pair, that is independent of the concentration waveform carrying syntactic information. The semantic and the syntactic information are multiplicatively coupled at the receptor level, leading to a confounding representation [4, 5] that can not be separated by single unit physiology recordings. To separate the two, characterizations of OSNs or PNs responses across channels are required. As this has not been possible due to limitations of the recording hardware, we sought a computational and theoretical approach to study the decoupling of odorant semantics and syntax in the fruit fly early olfactory system.

We first observed that single-channel physiology recordings at the output of the Antenna Lobe [35] exhibit *concentration-invariance* and *contrast-boosting* properties, indicating a time-dependent decoupling of the odorant object identity from the concentration waveform while responding strongly to odorant concentration onset and offset in transient states [40, 41]. Structurally, both the adult [31] and larva [32] *Drosophila* Antennal Lobes are extensively innervated by a diverse set of Local Neurons [3, 6, 14, 18, 23–30, 42–49] characterized by their connectivity patterns and neurotransmitter profiles.

For temporal processing of odors along a single channel, we modeled the presynaptic and postsynaptic (to OSN-PN synapses) LN pathways as *temporal* differential Divisive Normalization Processors, and demonstrated that the inhibition from Pre-LNs supports *concentration-invariance* in PN responses, while Post-eLNs and Post-iLNs enhance the *contrast-boosting*. Extending the differential DNP model to a spatio-temporal differential DNP model, we showed that the panglomerular inhibition from Pre-LNs is also able to produce concentration-invariant PN responses both across time and across channels.

We then hypothesized that the population PN responses, in particular the concentration-invariant components, *recovers* the odorant semantics (identity). We showed that the panglomerular inhibition from Pre-LNs can recover odorant object identity in a concentration-invariant manner, while the Post-eLNs and Post-iLNs signal onset and offset timing information of odorant stimuli in a time-synchronized manner. The ability to recover identities of the odorant objects supports disambiguation between different odorant stimuli as previously observed by studies investigating a small subset of glomeruli [15, 16, 50–52]. Additionally, from a spike-processing perspective, we proposed that the odorant semantics is mapped into stable attractors of the PN spiking neurons, while the semantic timing leads to transient traversal away from and back to these stable attractors in the phase space.

From a robustness standpoint, we showed that the odorant object identity recovery enabled by the Pre-LN pathway is robust across odorant object identity and concentration waveform profiles. Additionally, the Post-eLNs and Post-iLNs act as reliable event detectors capturing odorant onset and offset timing information across concentration waveform profiles. As the onset/offset events are tightly integrated with the semantics of an odorant object, we coined the term *semantic timing* to describe such information streams. The robustness of the Pre- and Post-LN pathway responses, and the separation of odorant object identity and concentration waveform profile along these pathways strongly suggest that the functional logic of the AL is that of an *ON-OFF odorant object identity recovery processor*. Within the context of *semantic* vs *syntactic* odorant information, *semantic* information of the odorant stimulus can be directly read-out from the PN responses, and the read-out does not depend on the *syntactic* information of the concentration waveform. We note that the temporal integral of the concentration contrast equates the absolute concentration amplitude in log-space. Thus, the absolute odorant concentration amplitude is also provided by the response of the PNs.

While not discussed in detail in this paper, we also performed extensive computational experiments to investigate the functional significance of panglomerular presynaptic inhibition

in the AL circuit (see also [53] for more extensive comparative analyses of the entire AL circuit architectures). Specifically, in Section 4.4 in the Methods section, we compared syntactic and semantic processing of different variations of the Pre-LN pathway. For example, by changing the Pre-LN input from OSN spike train, we compared spatio-temporal differential DNP models between *feedforward* inhibition and *feedback* inhibition. We found that both configurations are able to robustly recover the odorant object identity. In contrast to the robustness of the feedforward/feedback configurations, however, changing the input to Pre-LN from panglomerular to uni-glomerular significantly deteriorated the odorant semantic recovery performance. Our experiments suggested that the multi-channel AL circuit architecture is ideally suited to solving the odor information processing problem at the level of the PNs, consistent with the our intuition afforded by the theoretical analysis of spatio-temporal differential DNPs in Section 2.2.

A partial validation of the connectivity structure of the graphs underlying the model circuits of the current work can be obtained from a detailed mapping the LN pathways into the available connectomics datasets [31, 32]. For the three LN pathways modeled here, neurons corresponding to the Global Pre-LN have been reported [32, 54, 55] (e.g., Broad Trio in larval AL [32]), and excitatory Post-LNs have also been previously reported in [33]. However, accurate mapping of our model to the "wetware" of biological brains would require comparative analyses of multiple connectomics dataset with synaptic information including neurotransmitter profile—a task for which the current work provides a hypothetical scaffolding. How to identify LNs in the massive feedforward/feedback circuit of the AL that sense the presence/absence of odorant objects remains a major challenge for years to come. Additionally, we note that, for analytical tractability, the AL circuits models do not capture the full complexity of the Antennal Lobe. For example, the AL processing paradigm studied here assumes that the population response of the uni-glomerular PNs recover the odorant object identity, without multi-glomerular PNs. Also, due to the lack of comprehensive gap junction information, our AL circuit model does not include gap junctions. Furthermore, the LN cell-types are assumed to be either pan-glomerular or uni-glomerular, while many LNs in the AL innervate a subset of the glomeruli. Nevertheless, the circuit models presented herein can be readily extended to incorporate these neuron types, and we are actively working on bridging the gap between computational AL models and biological data [54, 55].

Since the odorant object identity is embedded into the PN population response, experimental verification of our models require, e.g., simultaneous recordings of uni-glomerular PNs projecting to a large number of (if not all) glomeruli. Biological verification of the multi-channel model, therefore, calls for massively parallel recordings and a systematic characterization of populations of PN responses of the AL.

## 4 Materials and methods

In Section 4.1, we describe the modeling methodology and circuit component details of single-channel and multi-channel AL circuits. In Section 4.2 we formally define concentration contrast and describe algorithms for decomposing the PN PSTH and the synaptic current injected into PNs as a piecewise constant signal component and a contrast-boosting transient component. Based on the onset/offset timing information determined from the transient component, the amplitude of the piecewise constant component is found as the average value between ON/OFF events. In Section 4.3 we define the optimization approach and algorithms for evaluating the functional logic of the AL circuits. In Section 4.4 we describe additional comparative analyses between different configurations of AL circuit architectures that highlight the functional significance of the LN pathways.

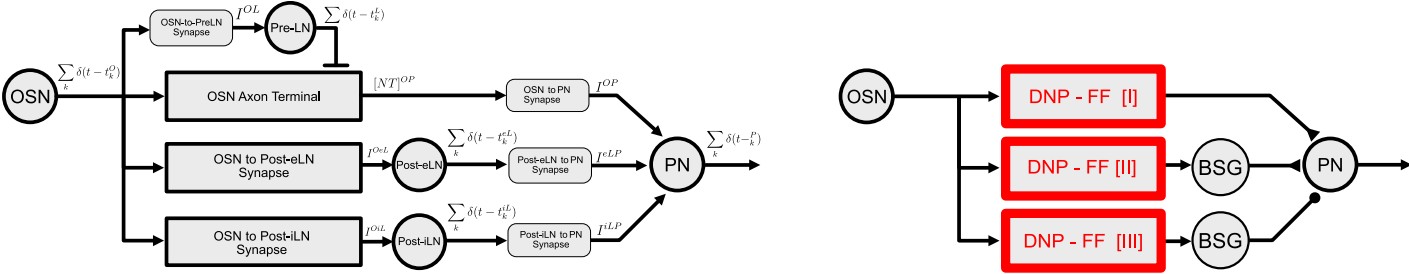

**Fig 10. (left) Example single-channel AL circuit and (right) the associated model DNP circuit.** Refer to Table 1 for details on the circuit component parameterizations and the correspondence between circuit component models and the differential DNP models of Eq (3).

## 4.1 Differential DNP models of the architecture of the antennal lobe

As described in Section 2.1, all LN pathways (Pre-LN, Post-eLN, Post-iLN) in the Antennal Lobe can be characterized, drawing on Eqs (1), (3) and (4), as differential DNPs in feedforward and feedback configurations. In what follows we detail the models of single-channel and multi-channel Antennal Lobe circuits, and their relations to temporal and spatio-temporal differential DNPs. In particular, we present an example single-channel feedforward (Fig 10) and an example multi-channel feedforward (Fig 11) AL circuit, and their corresponding differential DNP-FFs. Later in Section 4.4 we compare configurations of the circuit architecture and their impact on the processing of syntactic and semantic odorant information. Note that all circuit components and their corresponding models are parameterized by a set of free

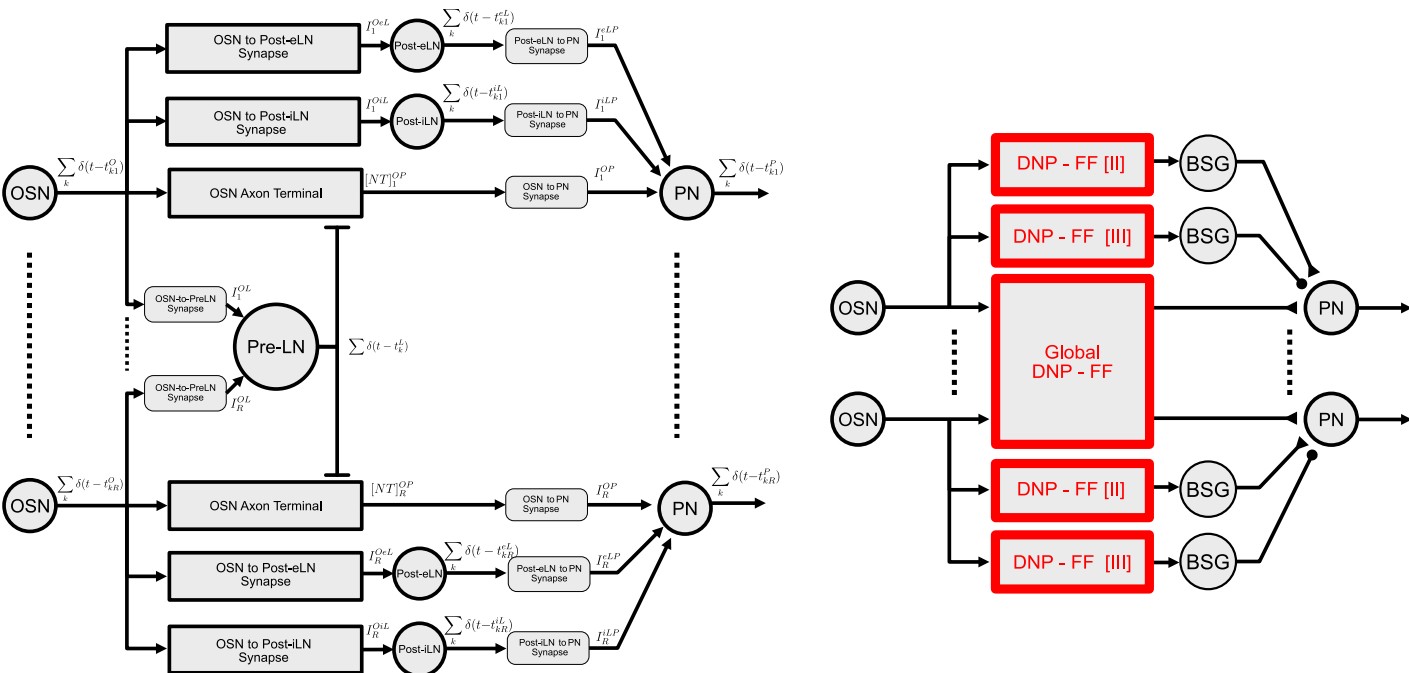

**Fig 11. (left) Example multi-channel AL circuit with feedforward Pre-LN pathway across glomeruli and (right) the associated model DNP circuit.** The multi-channel AL circuit has *R* channels, receiving input from OSNs expressing *R* receptor types. Shown above on the left are channels 1 and *R*. The Pre-LN pathway receives spiking input from OSNs across channels and outputs a single Pre-LN spike train. As opposed to the cross-channel integration of the Pre-LN pathway, the Post-eLN and Post-iLN pathways are repeated for each channel and only perform computation local to their corresponding glomerulus. The same distinction of global vs. local computations are shown for the model differential DNP circuit on the right. Here the multi-channel Pre-LN pathway is modeled as a *Global DNP-FF*. Note that the Post-eLN and Post-iLN pathways correspond to the same DNP-FF[II] and DNP-FF[III] models as in the single-channel AL circuit in Fig 10. Refer to Table 2 for model details.

parameters $\boldsymbol{\theta} = (\boldsymbol{\theta}_1, \boldsymbol{\theta}_2, \boldsymbol{\theta}_3)$, where $\boldsymbol{\theta}_1, \boldsymbol{\theta}_2, \boldsymbol{\theta}_3$ correspond to free parameters of the circuits components in the Pre-LN, Post-eLN and Post-iLN pathways, respectively. The entries of free parameters in $\boldsymbol{\theta}$ are listed in detail in **S2 Table in** S1 Appendix.

**Single-channel antennal lobe circuit.** In Fig 10(left), we expanded upon the single-channel AL circuit model shown in Fig 3 in Section 2.1. Each circuit component of the AL circuit in Fig 10(left) is described by a set of differential equations detailed in Table 1. Additionally, as shown in Fig 10(right), the three LN pathways are modeled as three differential DNPs (DNP-FF [I-III]). For example, the DNP-FF [I] component in Fig 10(right) describes the Pre-LN pathway that maps the OSN spike train $\sum_k \delta(t - t_k^O)$ into the OSN Axon-Terminal neurotransmitter concentration $[NT]^{OP}$, listed in Table 1(1–3). The operators $\mathcal{T}_3, \mathcal{T}_4$ of the DNP-FF [I] are explicitly defined in the same rows of Table 1(1–3) as 1) an identity operator that maps the OSN spike train into itself and 2) a transformation operator that maps the OSN spike train into the Pre-LN BSG spike train. Similarly in Table 1(4–5) and Table 1(6–7), the components

**Table 1. Model description of the AL circuit in Fig 10(left), and its correspondence with the DNP circuit in Fig 10(right).** Note that for notational simplicity the free parameter values for $\boldsymbol{\theta}$ are omitted. Indices under the *Model* column refer to the following detailed models: (1) OSN to Pre-LN Synapse, (2) Pre-LN BSG, (3) OSN Axon-Terminal, (4) OSN to Post-eLN Synapse, (5) Post-eLN BSG, (6) OSN to Post-iLN Synapse, (7) Post-iLN BSG, (8) OSN to PN Synapse, (9) Post-eLN to PN Synapse, (10) Post-iLN to PN Synapse, (11) PN BSG. The differential equations for each model are detailed under the *Equation* column, with the corresponding parameters summarized under the *Parameters* column. Note that under the *Parameters* column, values for a number of fixed parameters are shown. The circuit has a total of 23 free parameters denoted by the vector $\boldsymbol{\theta}$. For models that correspond to the three differential DNP components shown in Fig 10(right), the correspondences are made explicit on the right side of rows (1–3), (4–5), (6–7) for, respectively, DNP-FF[I], DNP-FF[II] and DNP-FF[III].

| Model | Parameters | Equation | Differential DNP Operators | |
|---|---|---|---|---|
| (1) | $[\alpha_1, \beta_1, \bar{g}_{max}^{OL}]$ | $\frac{d}{dt} x^{OL} = \alpha_1 \cdot \sum_k \delta(t - t_k^O) \cdot (1 - x^{OL}) - \beta_1 \cdot x^{OL}$ | $\mathcal{T}_3 : \sum_k \delta(t - t_k^O) \mapsto \sum_k \delta(t - t_k^O)$ | DNP-FF [I] |
| | | $I^{OL} = \bar{g}_{max}^{OL} \cdot x^{OL} \cdot (V^L - E^{OL})$ | $\mathcal{T}_4 : \sum_k \delta(t - t_k^O) \mapsto \sum_k \delta(t - t_k^L)$ | |
| (2) | $\sigma^L = 0$ | $\sum_k \delta(t - t_k^L) \leftarrow \text{NoisyConnorStevens}(I^{OL}; \sigma^L)$ | | |
| (3) | $[\alpha_2, \beta_2, \kappa_2, \overline{[NT]}_{max}]$ | $\frac{d}{dt} x^{AxT} = \alpha_2 \cdot \sum_k \delta(t - t_k^O) \cdot (1 - x^{AxT}) - \beta_2 \cdot x^{AxT} - \kappa_2 \cdot \sum_k \delta(t - t_k^L) \cdot x^{AxT}$ | Differential DNP Eq (3) | |
| | | $[NT]^{OP} = \overline{[NT]}_{max} \cdot x^{AxT}$ | | |
| (4) | $[\alpha_3, \beta_3, \kappa_3, \alpha_4, \beta_4, \alpha_5, \beta_5, \bar{g}_{max}^{OeL}]$ | $\frac{d}{dt} x^{OeL} = \alpha_3 \cdot x_2^{OeL} \cdot (1 - x^{OeL}) - \beta_3 \cdot x^{OeL} - \kappa_3 x_3^{OeL} \cdot x^{OeL}$ | Differential DNP Eq (3) | DNP-FF [II] |
| | | $\frac{d}{dt} x_2^{OeL} = \alpha_4 \cdot \sum_k \delta(t - t_k^O) \cdot (1 - x_2^{OeL}) - \beta_4 \cdot x_2^{OeL}$ | $\mathcal{T}_3 : \sum_k \delta(t - t_k^O) \mapsto x_2^{OeL}$ | |
| | | $\frac{d}{dt} x_3^{OeL} = \alpha_5 \cdot \sum_k \delta(t - t_k^O) \cdot (1 - x_3^{OeL}) - \beta_5 \cdot x_3^{OeL}$ | $\mathcal{T}_4 : \sum_k \delta(t - t_k^O) \mapsto x_3^{OeL}$ | |
| | | $I^{OeL} = \bar{g}_{max}^{OeL} \cdot x^{OeL} \cdot (V^{eL} - E^{OeL})$ | | |
| (5) | $\sigma^{eL} = 0$ | $\sum_k \delta(t - t_k^{eL}) \leftarrow \text{NoisyConnorStevens}(I^{OeL}; \sigma^{eL})$ | | |
| (6) | $[\alpha_6, \beta_6, \kappa_6, \alpha_7, \beta_7, \alpha_8, \beta_8, \bar{g}_{max}^{OiL}]$ | $\frac{d}{dt} x^{OiL} = \alpha_6 \cdot x_2^{OiL} \cdot (1 - x^{OiL}) - \beta_6 \cdot x^{OiL} - \kappa_6 x_3^{OiL} \cdot x^{OiL}$ | Differential DNP Eq (3) | DNP-FF [III] |
| | | $\frac{d}{dt} x_2^{OiL} = \alpha_7 \cdot \sum_k \delta(t - t_k^O) \cdot (1 - x_2^{OiL}) - \beta_7 \cdot x_2^{OiL}$ | $\mathcal{T}_3 : \sum_k \delta(t - t_k^O) \mapsto x_2^{OiL}$ | |
| | | $\frac{d}{dt} x_3^{OiL} = \alpha_8 \cdot \sum_k \delta(t - t_k^O) \cdot (1 - x_3^{OiL}) - \beta_8 \cdot x_3^{OiL}$ | $\mathcal{T}_4 : \sum_k \delta(t - t_k^O) \mapsto x_3^{OiL}$ | |
| | | $I^{OiL} = \bar{g}_{max}^{OiL} \cdot x^{OiL} \cdot (V^{iL} - E^{OiL})$ | | |
| (7) | $\sigma^{iL} = 0$ | $\sum_k \delta(t - t_k^{iL}) \leftarrow \text{NoisyConnorStevens}(I^{OiL}; \sigma^{iL})$ | | |
| (8) | $[\alpha_9 = 1, \beta_9 = 100, \bar{g}_{max}^{OP} = 100]$ | $\frac{d}{dt} x^{OP} = \alpha_9 \cdot [NT]^{OP} \cdot (1 - x^{OP}) - \beta_9 \cdot x^{OP}$ | | |
| | | $I^{OP} = \bar{g}_{max}^{OP} \cdot x^{OP} \cdot (V^P - E^{OP})$ | | |
| (9) | $[\alpha_{10} = 1, \beta_{10} = 100, \bar{g}_{max}^{eLP} = 100]$ | $\frac{d}{dt} x^{eLP} = \alpha_{10} \cdot \sum_k \delta(t - t_k^{eL}) \cdot (1 - x^{eLP}) - \beta_{10} \cdot x^{eLP}$ | | |
| | | $I^{eLP} = \bar{g}_{max}^{eLP} \cdot x^{eLP} \cdot (V^P - E^{eLP})$ | | |
| (10) | $[\alpha_{11} = 1, \beta_{11} = 100, \bar{g}_{max}^{iLP} = 100]$ | $\frac{d}{dt} x^{iLP} = \alpha_{11} \cdot \sum_k \delta(t - t_k^{iL}) \cdot (1 - x^{iLP}) - \beta_{11} \cdot x^{iLP}$ | | |
| | | $I^{iLP} = \bar{g}_{max}^{iLP} \cdot x^{iLP} \cdot (V^P - E^{iLP})$ | | |
| (11) | $\sigma^P = 0.0013849367$ | $\sum_k \delta(t - t_k^P) \leftarrow \text{NoisyConnorStevens}(I^P = (I^{OP} + I^{eLP} + I^{iLP}); \sigma^P)$ | | |

**Table 2. Model description of Fig 11(left), and its correspondence with the DNP circuit in Fig 11(right).** Note that for notational simplicity the parameter values for $\theta$ are omitted. Indices under the *Model* column refer to the following detailed models: (1) OSN to Pre-LN Synapse in the *r*-th channel, (2) Pre-LN BSG, (3) OSN Axon-Terminal in the *r*-th channel. Note that only components in the Global DNP-FF pathway are specified as the other circuit components are the same as in Fig 10(left) for each of the *R* channels. The differential equations for each model are detailed under the *Equation* column, with the corresponding parameters summarized under the *Parameters* column. Note that under the *Parameters* column, values for fixed parameters are shown. The circuit has a total of 23 free parameters denoted by vector $\theta$, the same as the single-channel AL circuit in Fig 10. The correspondence between the Pre-LN pathway and the Global DNP-FF model is made explicit on the right side of rows (1–3).

| Model | Parameters | Equation | Differential DNP Operators | |
|---|---|---|---|---|
| (1) | $[\alpha_{14}, \beta_{14}, \bar{g}_{max}^{OL}]$ | $\frac{d}{dt}x_r^{OL} = \alpha_{14} \cdot \sum_k \delta(t - t_{kr}^O) \cdot (1 - x^{OL}) - \beta_{14} \cdot x_r^{OL}$ | $\mathcal{T}_7 : \sum_k \delta(t - t_{kr}^O) \mapsto \sum_k \delta(t - t_{kr}^O)$ | Global DNP-FF |
| | | $I_r^{OL} = \bar{g}_{max}^{OL} \cdot x_r^{OL} \cdot (V^L - E^{OL})$ | $\mathcal{T}_8 : \{\sum_k \delta(t - t_{kr}^O)\}_{r=1}^R \mapsto \sum_k \delta(t - t_k^L)$ | |
| (2) | $\sigma^L = 0$ | $\sum_k \delta(t - t_k^L) \leftarrow \text{NoisyConnorStevens}(\sum_{r=1}^R I_r^{OL}; \sigma^L)$ | | |
| (3) | $[\alpha_{15}, \beta_{15}, \kappa_{15}, \overline{[NT]}_{max}]$ | $\frac{d}{dt}x_r^{AxT} = \alpha_{15} \cdot \sum_k \delta(t - t_{kr}^O) \cdot (1 - x_r^{AxT}) - \beta_{15} \cdot x_r^{AxT} - \kappa_{15} \cdot \sum_k \delta(t - t_k^L) \cdot x_r^{AxT}$ | Differential DNP Eq (4) | |
| | | $[NT]_r^{OP} = \overline{[NT]}_{max} \cdot x_r^{AxT}$ | | |

DNP-FF [II] and DNP-FF [III] respectively model the Post-eLN and Post-iLN pathways. Note that DNP-FF [II] and DNP-FF [III] in Table 1 are described by the same set differential equations. Only the parameter values may differ.

**Multi-channel antennal lobe circuit.** In Fig 11, we expanded upon the multi-channel AL circuit shown in Fig 4 in Section 2.1. As shown in Fig 11(left), the Post-eLN and Post-iLN pathways are local to each channel (glomerulus), corresponding to the DNP-FF[II] and DNP-FF[III] components for each channel as shown in Fig 11(right). The circuit components of the Post-eLN and Post-iLN pathways are fully specified by their single-channel counterparts in the previous section and are therefore omitted in Table 2 for brevity. Consequently, only the multi-channel Pre-LN pathway is specified in Table 2, that corresponds to the *Global DNP-FF* model shown in Fig 11(right). The Global DNP-FF model is described by the *Global Feedforward* Eq (4). As shown in Table 2(1–3), the operators $\mathcal{T}_7$ map the OSN spike train in each channel into itself, and the operator $\mathcal{T}_8$ maps the OSN spike trains along all channels into a single Pre-LN output spike train.

## 4.2 Syntactic and Semantic Odor Information Representation by PN BSGs

In the current section, we defined the syntactic and semantic information of the odorant stimuli, and their corresponding representation at the level of the PN BSGs.

**Characterization of the syntactic and semantic information of odorant stimuli.** For the *n*-th OSN expressing receptor type *r* that is binding to the odorant indexed *o* with concentration waveform $u = u(t)$, $t \geq 0$, the odorant object stimulus is defined as [2]

$$([\mathbf{b}]_{ron} \cdot u(t), [\mathbf{d}]_{ron}), \quad r = 1, \ldots, R, \quad n = 1, \ldots, N, \quad o \in \mathcal{O},$$

where *R* is the total number of receptors (51 in adult *Drosophila*), *N* is the number of OSNs expressing the same receptor type, $\mathcal{O}$ is the set of all odorants ($|\mathcal{O}| = 110$ in the DoOR2.0 [10] dataset). Assuming that all OSNs expressing the same receptor type have the same binding/dissociation rates for a given odorant and $N = 1$ (single OSN per receptor type), we can drop the subscript *n* in the definition above and rewrite the odorant stimulus in vector form as

$$(\mathbf{b}_o \cdot u(t), \mathbf{d}_o),$$

where $\mathbf{b}_o = [[\mathbf{b}]_{1o}, \ldots, [\mathbf{b}]_{Ro}]^T$ and $\mathbf{d}_o = [[\mathbf{d}]_{1o}, \ldots, [\mathbf{d}]_{Ro}]^T$. In the odorant stimulus model above, the *syntactic* information is represented by the concentration waveform $u(t)$ while the *semantic* information is represented by the odorant binding/dissociation rates ($\mathbf{b}_o, \mathbf{d}_o$). In another word, the semantic information of an odorant is identity- but not concentration-dependent.

In what follows we define the *concentration contrast* of the odorant concentration waveform as:

$$\text{Contrast}[u](t) = \frac{du/dt}{\epsilon + u(t)} = \frac{d}{dt} \log(\epsilon + u(t)), \tag{7}$$

where $\epsilon$ is a small bias term that avoids division by zero. The contrast of the Acetone staircase concentration waveform depicted in Fig 2A(bottom) was obtained for $\epsilon = 1$. Note that the temporal integral of the concentration contrast in Eq (7) amounts to the absolute concentration value in the log-space.

**Characterization of odorant semantics & semantic timing with PN BSGs.** At the output stage of the Antennal Lobe circuit, the synaptic input currents (in $[\mu A]$) are described by the vector parametrized by $\boldsymbol{\theta}$

$$\boldsymbol{I}_o^P(t; \boldsymbol{\theta}) = [I_{1o}^P(t; \boldsymbol{\theta}), \ldots, I_{ro}^P(t; \boldsymbol{\theta}), \ldots, I_{Ro}^P(t; \boldsymbol{\theta})]^T,$$

whose entries are encoded by PN BSGs modeled as Connor-Stevens point neurons (Table 1 (row 11)). At a fixed input current amplitude, PN BSGs exhibit stable periodic oscillations, which in the phase-space are represented as the limit cycles depicted in Fig 6A and 6C [38]. Specifically, in Fig 6A and 6C, a collection of limit cycles are shown for PN BSGs with different input current amplitudes, where the (x, y) coordinates are labeled by two of the state variables of the PN BSG ($V$, $n$), and the z-direction corresponds to the amplitude of the input current (in $\mu A$). Note that as the input current increases, the diameter of the limit cycles decreases, corresponding to a higher spike rate. The PN BSG input currents shown in Fig 3B(left) are the superposition of a concentration-invariant component and on/off contrast-boosting components. An example of the phase-space dynamics of the PN BSG is shown in Fig 7A[iii]. Here the BSG oscillates around a *stable attractor* (Fig 7A[iii, black]) and traverses to higher (in z-direction in phase-space) or lower limit cycles in response to transient input current changes. Critically, this suggests that the PN BSGs' ON/OFF timing events across glomeruli are differently localized (in time) when compared to the representation of odorant semantics. Note that the odorant semantics is represented by the set of stable attractors in Fig 6A and 6C, and the semantic timing of an odorant object is represented as the onset/offset events that lead to dynamics of the limit cycles away from the stable attractors.

Quantitatively, the phase-space representation of the PN BSG dynamics is closely related to the PN PSTH. From the phase-space representation, the spike train generated by the PN BSG $\sum_k \delta(t - t_{kr}^P; \boldsymbol{\theta})$, $r = 1, \ldots, R$, (denoted as $\sum_k \delta(t - t_k^P)$ in Table 1(row 11) for single-channel PN output, where subscript $r$ was omitted for notational simplicity) provides the timing of the voltage reaching the maximum amplitude value. From the spike train, the Peri-Stimulus Time Histogram (PSTH) of the PN BSGs (in $[Hz]$) parametrized by $\boldsymbol{\theta}$

$$\boldsymbol{\lambda}_o^P(t; \boldsymbol{\theta}) = [\lambda_{1o}^P(t; \boldsymbol{\theta}), \ldots, \lambda_{ro}^P(t; \boldsymbol{\theta}), \ldots, \lambda_{Ro}^P(t; \boldsymbol{\theta})]^T$$

can then be estimated using a window size of 200 ms with 100 ms overlap. Note that the DM4 PN physiology recording data $\hat{\lambda}_{ro}^P(t)$ is obtained by using the same PSTH estimation procedure. Moreover, the synaptic current $\boldsymbol{I}_o^P(t; \boldsymbol{\theta})$ (Table 1(row 11)) and PN PSTH $\boldsymbol{\lambda}_o^P(t; \boldsymbol{\theta})$ are related by the increasing Frequency-Current function $f$ of the PN BSG model (e.g., Conner-Stevens point neuron model) and

$$f(\boldsymbol{I}_o^P(t; \boldsymbol{\theta})) = \boldsymbol{\lambda}_o^P(t; \boldsymbol{\theta}).$$

The PN input synaptic current is then obtained from the model output PSTH by the

inversion ($I_o^P(t; \boldsymbol{\theta}) = f^{-1}(\lambda_o^P(t; \boldsymbol{\theta}))$), and similarly from DM4 physiology recording as $\hat{I}_{ro}^P(t) = f^{-1}(\hat{\lambda}_{ro}^P(t))$, $r = 1, \ldots, R$.

Procedurally, PN PSTH and the injected synaptic current can be decomposed in the phase space into a stable attractor and transient components in the following manner (see Fig 2C (bottom) for an example):

1. *Extracting the odorant concentration waveform jump times*: we first evaluated the odorant concentration contrast using Eq (7) with $\varepsilon = 1$, and stored the timing of the positive and negative peak values as *jump times*.

2. *Semantic Information Representation*: we then created a piecewise constant signal component of the PN PSTH with transition times specified by the jump times, and the amplitude specified by the average spike rate during the 500 milliseconds prior to each jump time. Using the same procedure, a piecewise constant component can be extracted from the overall synaptic current injected into the PN. As discussed above, this piecewise constant component is concentration-invariant, the PN BSG oscillates around its stable attractor. As this signal component of the PN input synaptic current is attributed to the OSN Axon-Terminal to PN synaptic current along the Pre-LN pathway, we simply denoted the model response as $I_{ro}^{OP}(t; \boldsymbol{\theta})$ as in Table 1(row 8), where the estimate obtained from the physiology recordings is denoted by $\hat{I}_{ro}^{OP}(t)$.

3. *Semantic Timing Information Representation*: we created the transient response as the residual obtained by subtracting the piecewise constant signal component obtained in the previous step from the overall response. The transient component of the input synaptic current into the PN BSG is further decomposed into the ON and OFF components, corresponding to the Post-eLN and Post-iLN pathways, denoted as

$$I_{ro}^{eLP}(t; \boldsymbol{\theta}) = [I_{ro}^P(t; \boldsymbol{\theta}) - I_{ro}^{OP}(t; \boldsymbol{\theta})]_+, \quad I_{ro}^{iLP}(t; \boldsymbol{\theta}) = [I_{ro}^P(t; \boldsymbol{\theta}) - I_{ro}^{OP}(t; \boldsymbol{\theta})]_-,$$

respectively, as in Table 1(row 9, 10). Here $[\cdot]_+, [\cdot]_-$ are positive/negative rectification operators. The same decomposition applies for the physiology output $\hat{I}_{ro}^{eLP}(t), \hat{I}_{ro}^{iLP}(t)$.

## 4.3 Methods of Optimization of the Antennal Lobe Circuit

**Metrics.** The odorant concentration waveforms and output PN PSTHs considered in the current work are assumed to be in the space of square-integrable functions with temporal support $[0, T]$ (not necessarily bandlimited). For such signals, we respectively use the $L_2$ norm and the inner-product

$$\|u(t)\|^2 = \frac{1}{T} \int_0^T |u(t)|^2 dt, \quad \langle u, v \rangle = \frac{1}{T} \int_0^T u(t)v(t)dt,$$

where the $1/T$ normalization is introduced to simplify the comparison between two signals $u$ and $v$ with different temporal support. For spatio-temporal square-integrable signals $\mathbf{u}(t) = [u_1(t), u_2(t), \ldots, u_N(t)]^T$ with temporal support $[0, T]$, the $L_2$ norm and the inner-product is similarly defined as

$$\|\mathbf{u}(t)\|^2 = \frac{1}{T} \int_0^T \sum_{i=1}^N |u_i(t)|^2 dt, \quad \langle \mathbf{u}, \mathbf{v} \rangle = \frac{1}{T} \int_0^T \sum_{i=1}^N u_i(t)v_i(t)dt.$$

For physiology data $\hat{u}(t)$ and model output $u(t)$, the Signal-to-Noise Ratio (SNR) is given (in

deciBel) as

$$SNR(\hat{u}, \hat{u} - u) = 20 \log_{10} \left[ \frac{\|\hat{u}(t)\|}{\|\hat{u}(t) - u(t)\|} \right].$$

A scale-invariant pseudo-metric used in the current work is the angular distance. For temporal signals $u = u(t)$, $v = v(t)$, $t \geq 0$, this is defined as:

$$d_\angle(u, v) = \frac{2}{\pi} \cdot \arccos \left( \frac{\langle u, v \rangle}{\|u\| \|v\|} \right) \in [0, 1].$$

The definition extends readily to vector-valued functions $\mathbf{u} = \mathbf{u}(t)$, $\mathbf{v} = \mathbf{v}(t)$, $t \geq 0$:

$$d_\angle(\mathbf{u}, \mathbf{v}) = \frac{2}{\pi} \cdot \arccos \left( \frac{\langle \mathbf{u}, \mathbf{v} \rangle}{\|\mathbf{u}\| \|\mathbf{v}\|} \right) \in [0, 1],$$

where the norm and inner-product operators are defined above.

**Optimization objectives for AL circuit models.**   The parameters for both the single-channel and multi-channel AL circuit models are obtained by *independently* deriving the optimal solution of a different objective function set up for each of the three Local Neuron pathways (Pre-LN, Post-eLN and Post-iLN). For each of the three pathways, we optimized the angular distances between the model and physiology recordings of the synaptic currents injected by the OSN Axon-Terminal into the PN (for the Pre-LN pathway), the Post-eLN into the PN and the Post-iLN into the PN.

Before describing the details of the respective optimization problems, we note that for each of the three LN pathways, only a subset of the total number of free parameters (e.g., 23 in Table 1) affect the value of the objective function. Specifically, we further specify $\boldsymbol{\theta} = (\boldsymbol{\theta}_1, \boldsymbol{\theta}_2, \boldsymbol{\theta}_3)$, where $\boldsymbol{\theta}_1$, $\boldsymbol{\theta}_2$, $\boldsymbol{\theta}_3$ correspond to free parameters of the circuits components in the Pre-LN, Post-eLN and Post-iLN pathways respectively. See **S2 Table in** S1 Appendix for more details.

**Optimizing concentration-invariance of the pre-LN pathway of the single-channel AL Circuit Model.**   Refer to Fig 12A[ii,top] and Fig 12A[iii, top].

$$\min_{\boldsymbol{\theta}_1} \quad \underbrace{d_\angle(\hat{I}_{ro}^{OP}(\cdot), I_{ro}^{OP}(\cdot; \boldsymbol{\theta}_1))}_{J_{11}(\boldsymbol{\theta}_1)},$$

$$\text{s.t.} \quad \max_t x_{ro}^{AxT}(t) < 0.8 \tag{8}$$

where $\cdot$ is a placeholder for the time variable, $\hat{I}_{ro}^{OP}$ is the synaptic current injected by the Or59b OSN Axon-Terminal into the DM4 PN and $I_{ro}^{OP}(t; \boldsymbol{\theta}_1)$ is the model synaptic current. $\hat{I}_{ro}^{OP}$ was estimated from the DM4 PN physiology recording dataset [35] following the procedure described in section Section 4.2. The angular distance is minimized when the OSN Axon-Terminal synaptic current exhibits the same concentration-invariance property as the physiology data. To ensure that the concentration-invariance is achieved via the model structure, and it is not an artifact of model saturation, the constraint imposed on the maximum value of $x_{ro}^{AxT}$ above was introduced to cap the normalized neurotransmitter concentration of the OSN Axon-Terminal to 0.8—a simple heuristic that avoids model saturation. See above.

**Optimizing the encoding of ON event timing by the post-eLN pathway.**   Refer to Fig 12A[ii,middle] and Fig 12A[iii, middle].

$$\min_{\boldsymbol{\theta}_2} \quad \underbrace{d_\angle(\hat{I}_{ro}^{eLP}(\cdot), I_{ro}^{eLP}(\cdot; \boldsymbol{\theta}_2))}_{J_{12}(\boldsymbol{\theta}_2)}, \tag{9}$$

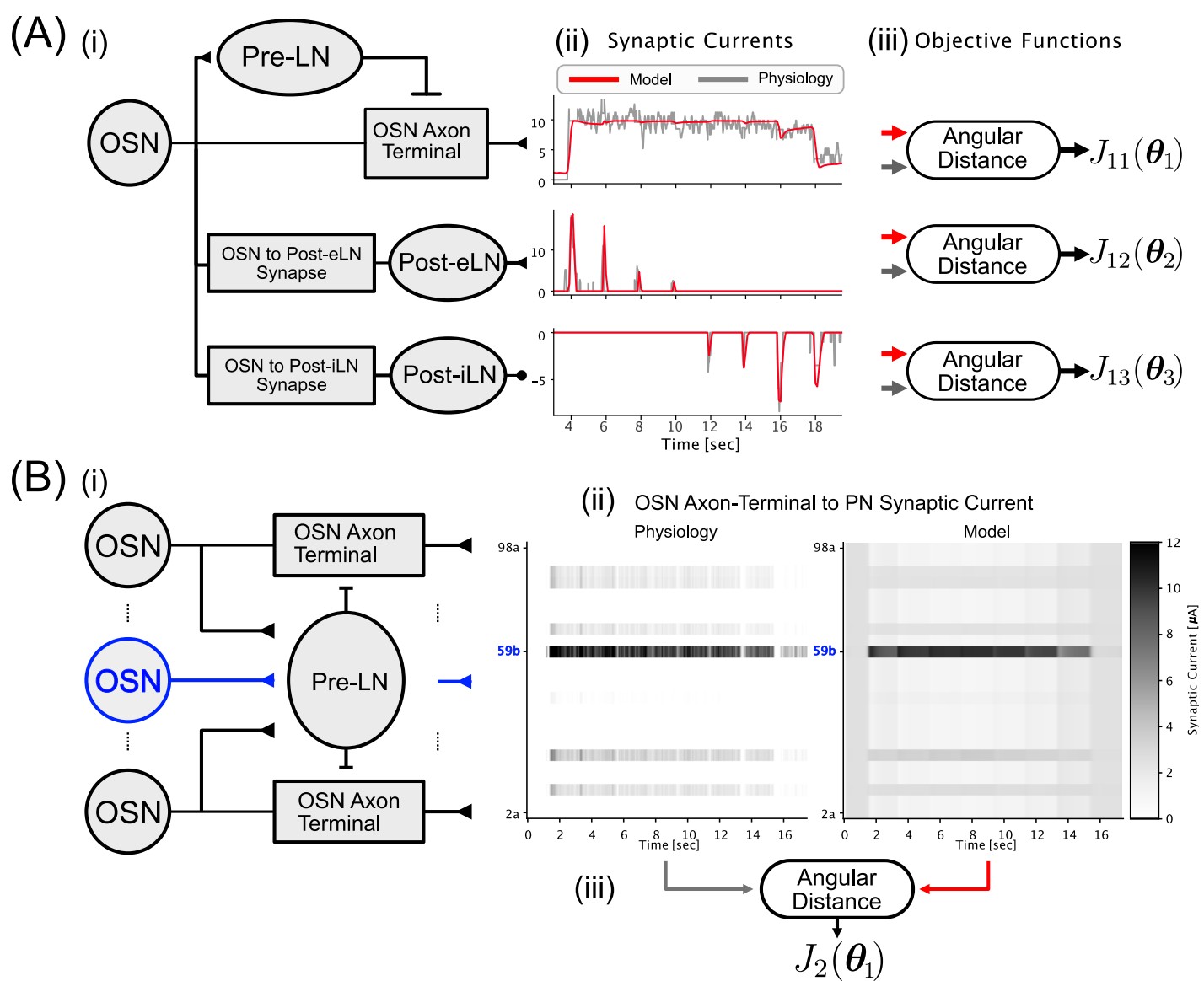

**Fig 12. Evaluation of the angular distance objective functions for the single- and multi-channel AL circuit models.** (A[i]) Single-channel AL circuit model. (A[ii]) Synaptic currents injected by the (top) OSN Axon-Terminal into the PN, (middle) Post-eLN into the PN and (bottom) Post-iLN into the PN. The synaptic current estimated from the physiology dataset [35] is shown in grey, and the synaptic current obtained from the circuit model is shown in red. (A[iii]) Objective functions of each of the three pathways in (A[ii]) respectively described by Eqs (8), (9) and (10). (B[i]) Pre-LN pathway in the multi-channel AL circuit model in Figs 4 and 11 and Table 2. The O59b OSN is highlighted in blue. (B[ii]) Synaptic current injected by the OSN Axon-Terminal into the PN shown as a heatmap. The synaptic current was estimated from (left) the physiology recording dataset in [35] and (right) the multi-channel AL circuit model. (B[iii]) Objective function for the multi-channel Pre-LN pathway.

where · is a placeholder for the time variable, $\hat{I}_{ro}^{eLP}$ is the synaptic current injected by the Post-eLN into the PN and $I_{ro}^{eLP}(t; \boldsymbol{\theta}_2)$ is the model PN synaptic current of the channel expressing the receptor type $r$ for odorant $o$. $\hat{I}_{ro}^{eLP}$ was estimated (see Section 4.2) from the DM4 PN physiology recording dataset in [35]. Note that we assumed that the parameter values of all circuit components along the Post-eLN pathway are shared across channels. As such, for both single-channel and multi-channel AL models, the optimization problem in Eq (9) focuses on capturing the transient dynamics of the Post-eLN to PN synaptic current as described above for the DM4

glomerulus. As discussed in Fig 8 and in **S4 Fig in** S1 Appendix, this leads to time-synchronized robust ON timing detection across channels, a temporal response feature observed in physiology recordings (see Section 2.1).

**Optimizing the encoding of OFF event timing by the post-iLN pathway.**   Refer to Fig 12A[ii,bottom] and Fig 12A[iii, bottom].

$$\min_{\boldsymbol{\theta}_3} \quad \underbrace{d_{\angle}(\hat{I}_{ro}^{iLP}(\cdot), I_{ro}^{iLP}(\cdot; \boldsymbol{\theta}_3))}_{J_{13}(\boldsymbol{\theta}_3)}, \tag{10}$$

where $\cdot$ is a placeholder for the time variable, $\hat{I}^{iLP}_{ro}$ is the Post-iLN to PN synaptic current and $I_{ro}^{iLP}(t; \boldsymbol{\theta}_3)$ is the model PN synaptic current. $\hat{I}^{iLP}_{ro}$ was estimated from our physiology recording dataset [35] following the procedure described in section Section 4.2. Note that, as in the Post-eLN pathway, we only optimize the parameters of Post-iLN pathway of DM4 glomerulus for both single-channel and multi-channel AL circuits, and assume that the resulting parameters are shared across all Post-iLN pathways of other glomeruli. As shown in Fig 8 and **S4 Fig in** S1 Appendix, this leads to time-synchronized robust OFF timing detection across channels, a temporal response feature observed in physiology recordings (see Section 2.1).

**Optimizing the recovery of odorant semantics by the multi-channel AL circuit model.** Refer to Fig 12B. The representation of odorant semantic information by the AL circuit is provided by the concentration-invariant and identity-dependent set of PN BSG stable attractors of each glomerulus. The spike rates of PN BSGs across glomeruli are proportional to the odorant affinity vector and, thereby, represent the odorant semantic information.

From an optimization standpoint, this can be obtained by extending the optimization problem in Eq (8) to the multi-channel case by measuring the angular distance, i.e.,

$$\min_{\boldsymbol{\theta}_1} \quad \underbrace{d_{\angle}(\hat{\mathbf{I}}_o^{OP}(\cdot), \mathbf{I}_o^{OP}(\cdot; \boldsymbol{\theta}_1))}_{J_2(\boldsymbol{\theta}_1)},$$
$$\text{s.t.} \quad \max_r \max_t x_{ro}^{AxT}(t) < 0.8 \tag{11}$$

where $\cdot$ is a placeholder for the time variable, $\hat{\mathbf{I}}_o^{OP} = [\hat{I}_{1o}^{OP}, \dots, \hat{I}_{Ro}^{OP}]^T$ is the vector-valued function of synaptic currents injected by the OSN Axon-Terminal into the PN and $\mathbf{I}_{ro}^{OP}(t; \boldsymbol{\theta}_1)$ is the model vector-valued function of the synaptic currents. Note that, by abuse of notation, we denoted the free parameter values of the Pre-LN pathway of the multi-channel AL circuit model as $\boldsymbol{\theta}_1$, with the understanding that the parameter values may differ from that of the single-channel AL circuit model in Eq (8). Similar to the inequality constraint in Eq (8), the constraint imposed above on the maximum value of $x_{ro}^{AxT}$ across both time and channels caps the normalized neurotransmitter concentration of the OSN Axon-Terminal to 0.8—a simple heuristic that avoids model saturation.

The proportionality between the affinity vector and the spike rates of PN BSGs is leveraged to estimate the OSN Axon-Terminal to PN synaptic current $\hat{\mathbf{I}}_o^{OP}$ for all channels given the DM4 PN physiology recording dataset [35]. Specifically, we estimated the multi-channel synaptic current $\hat{\mathbf{I}}_o^{OP}$ as follows:

- Estimate the multi-channel spatio-temporal PN PSTHs from the DM4 physiology recording dataset [35] using the affinity vector:

$$\hat{\boldsymbol{\lambda}}_o^P(t) = \left[ \underbrace{\frac{\mathbf{b}_{1o}/\mathbf{d}_{1o}}{\mathbf{b}_{ro}/\mathbf{d}_{ro}} \cdot \hat{\lambda}_{ro}^P(t)}_{\hat{\lambda}_{1o}^P(t)}, \ldots, \hat{\lambda}_{ro}^P(t), \ldots, \underbrace{\frac{\mathbf{b}_{Ro}/\mathbf{d}_{Ro}}{\mathbf{b}_{ro}/\mathbf{d}_{ro}} \cdot \hat{\lambda}_{ro}^P(t)}_{\hat{\lambda}_{Ro}^P(t)} \right]^T,$$

where $\mathbf{b}_{ro}/\mathbf{d}_{ro}$ is the affinity rate between Acetone and Or59b OSN;

- Estimate OSN Axon-Terminal to PN synaptic current $\hat{\mathbf{I}}_o^{OP}(t)$ given the Frequency-Current curve $f$ of the PN BSG model described in Section 4.2.

Note that by minimizing the overall angular distance computed across time and channels, we simultaneously enforce concentration-invariance and identity recovery of the PN population response.

**Two-step optimization procedure.** Optimization of the $\boldsymbol{\theta}$ parameters for all models presented herein follows a two-step algorithm: 1) random sampling in the parameter space using Latin Hypercube Sampling [56], 2) fine tuning using Differential Evolution [57].

**Step 1: Latin Hypercube Sampling (LHS)** [56] Latin Hypercube Sampling (LHS) is a method of generating random samples of parameter values from a multidimensional distribution. As opposed to uniform sampling along each dimension or grid sampling (which generates samples at regular intervals along each dimension), LHS picks random samples that satisfy the Latin Hypercube condition, i.e., one and only one sample from each axis-aligned hyperplane. For example, for LHS sampling in a 2-dimensional plane, samples [0, 1] and [0, 2] do not satisfy the condition as both samples share the same first dimension axis, while [3, 1] and [0, 2] satisfy the condition. However, enforcing the Latin Hypercube condition requires a pairwise comparison of all samples across all dimensions, a requirement that is computationally unfeasible in high dimensional spaces. We employed instead, based on the implementation in scikit-optimize [58], a simplified sampling procedure as follows:

1. Given a $D$-dimensional parameter space, we first generate $N$ initial samples by uniformly sampling each dimension independently, thereby creating a $N \times D$ dimensional matrix $X$, where each row corresponds to a sample in the parameter space. In our experiments, $N \approx 100,000$ and $D \approx 23$.

2. Randomly select $K$ rows ($K \approx 1,000$ used here) of the $X$ matrix, compute the pairwise correlation between the $K$ rows and save the average value of the correlations $\rho$.

3. Repeat step (1–2) for a total of $I$ iterations ($I \approx 10$ used here) and return the sample matrix $X$ that corresponds to the lowest average pairwise correlation $\rho$ between K rows.

Prior to performing the LHS procedure, we specified the range of feasible parameter values based on the stability constraint (see below) on the numerical simulation of the AL circuit model. By experimenting with different parameter values, we found that values larger than $10^6$ will result in numerical instability of the differential equation solvers for a time-step of 0.1 ms, that served as a heuristic upper bound on all model parameters. For a lower bound of parameter values, preliminary experiments showed that values below 1 resulted in PNs not producing any spikes even when the odorant concentration was high (e.g., 1,000 ppm). Therefore, the range of parameter values was chosen to be $[1, 10^6]$ for all parameters.

**Step 2: Differential Evolution** [57] For each LHS sample of the parameter space generated via step 1, the objective functions defined above (Eqs (8), (9), (10) and (11)) were evaluated to

obtain a coarse-scale perspective on the global cost function landscapes. To further reduce the objective function values, fine-tuning was then performed using Differential Evolution (DE) [57].

Differential Evolution (DE) is an evolutionary algorithm that seeks to solve a global optimization problem by iteratively improving candidate solutions within a hypercube parameter space. For the two-step optimization procedure, the hypercube is obtained by:

1. selecting the top 100 parameter sets found using LHS sampling (with the lowest objective function values), where each vector of parameter values is a sample in the $D$-dimensional parameter space,

2. computing the mean $\mu_d$ and the standard deviation $\sigma_d$ of all parameters for each dimension indexed by $d = \{1, \ldots, D\}$, and

3. creating a hypercube where each edge of the hypercube spans the range $[\mu_d - \sigma_d, \mu_d + \sigma_d]$.

The DE algorithm then randomly recombines and replaces candidate solutions sampled within the hypercube if they lead to a reduction of the optimization objective. The DE optimization procedure used in the current work is an accelerated implementation of scipy [59]. We performed all the optimization over the space of parameters $\theta_1, \theta_2, \theta_3$ for the Pre-LN, Post-eLN and Post-iLN pathways, respectively.

## 4.4 Comparative analysis of semantic odor information processing in the antennal lobe circuits

The single-channel and multi-channel AL circuits respectively described in Section 2.1 and Section 2.2 are fully specified by the both the circuit architecture and the parameters of the circuit components (see Fig 10 and Table 1 and, Fig 11 and Table 2, for example). The multi-channel AL circuit considered in Section 2.2 for example, consists of 1) multi-channel feedforward Pre-LN presynaptic inhibition, and 2) single-channel feedforward Post-eLN/Post-iLN postsynaptic excitation/inhibition. An alternative circuit architecture of the multi-channel AL circuit, for example, could be obtained by replacing the multi-channel feedforward Pre-LN presynaptic inhibition, with a multi-channel *feedback* Pre-LN presynaptic inhibition as shown in **S3 Fig in** S1 Appendix and **S4 Table in** S1 Appendix. The question therefore arises as to how the functional role of the LN pathways and their specific connectivity patterns can be evaluated by comparative analysis of different AL circuit architectures—a question that a computational approach to studying neural circuits is uniquely positioned to tackle.

For a fixed circuit architecture, the results presented in Section 2.1 and Section 2.2 are based on the parameters for the circuit components found via minimizing the objective functions $J_{11}, J_{12}, J_{13}, J_2$ over the set of circuit parameters $\theta$ as discussed in Section 4.3. In this section, we focus on comparing the ability for different circuit architectures to recover the odorant object identity. In particular, as shown in Fig 13A[i], we compare 5 different Pre-LN inhibition models: i) no inhibition, ii) feedback single-channel inhibition, iii) feedforward single-channel inhibition, iv) feedback multi-channel inhibition, v) feedforward multi-channel inhibition (as in Section 2.2). Note that for single-channel inhibition models [ii,iii], all OSN-to-PreLN synapses, Pre-LN BSG and Pre-LN-to-OSN synapses share the same parameter values across channels. As such, models [ii-v] have the same number of free parameters, and all models are optimized according to Eq (11). The angular distance matrices in Fig 13B extend the results shown in Fig 9A, displaying the angular distance between PN response and odorant affinity vectors for all 5 *optimized* circuits in Fig 13A.

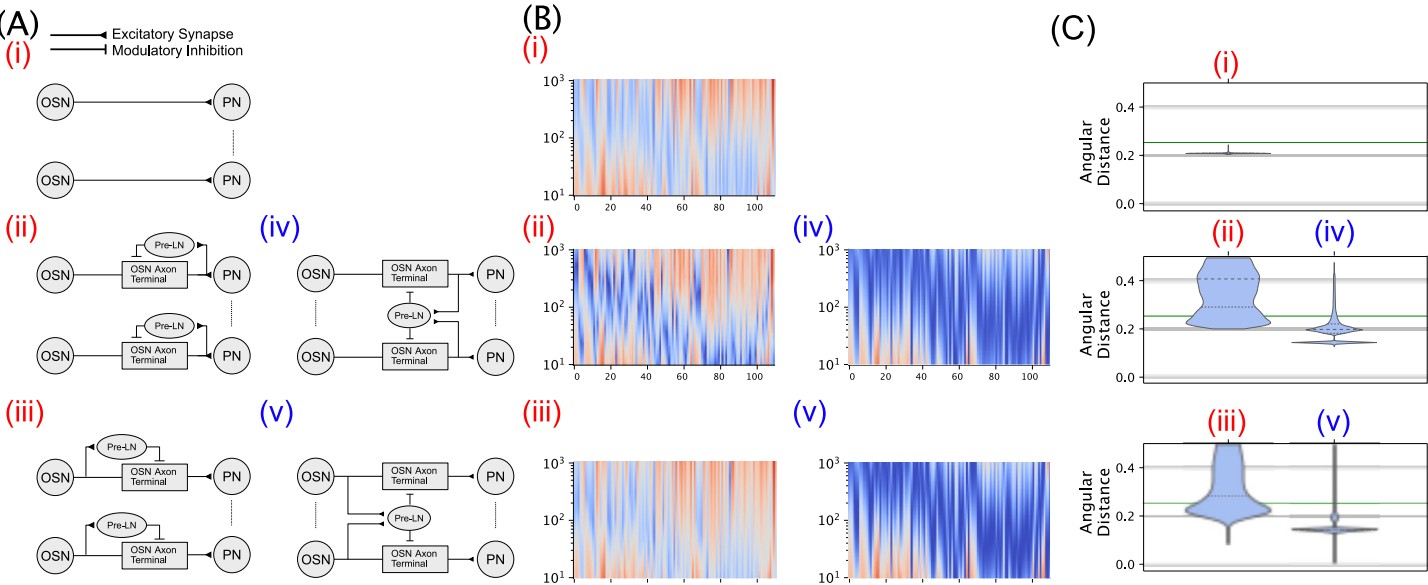

**Fig 13. Comparative analysis of AL circuit architectures for odor semantic information recovery.** (A) Circuit architectures of 5 different AL circuits: [i] no inhibition, [ii] feedback single-channel inhibition, [iii] feedforward single-channel inhibition, [iv] feedback multi-channel inhibition, [v] feedforward multi-channel inhibition. (B) Angular distance matrices between the PN population responses produced by the models [i-v] in (A) to 110 odorants with step concentration levels in the range of $10 \sim 1,000$ [*ppm*] (log scale). Refer to the colourbar on the right of Fig 9A for the scale of the angular distance from low (blue) to high (red). The x-axis corresponds to odorant indices, for a total of 110 odorants with known affinity rates. The y-axis corresponds to concentration levels across 2 orders of magnitude, from 10 ppm to 1000 ppm (log scale). See caption of Fig 9A for more details. (C) Empirical distributions of the objective function $J_2(\boldsymbol{\theta}_1)$ for randomly sampled parameter sets $\boldsymbol{\theta}_1$ for the 5 different AL circuits compared. Each violinplot along the x-axis corresponds to a different model in (A).

While the results in Fig 13B show that the global inhibition (model [iv, v]) result in far better odorant object identity recovery than the local inhibition models, such comparison is only limited to the locally optimal parameter values for each circuit architecture. Due to the nonconvex nature of the optimization problems in Section 4.3, the differences observed in Fig 13B may be attributed to the fact that each circuit architecture only reaches *local* optima, rather than *global* optima. To obtain a global perspective on the performance of odor information processing we compared the overall objective function landscapes of the 5 different AL circuit architectures in Fig 13A across different parameter set values. To obtain the objective function landscape, we randomly sampled a large number of (in excess of $10^6$) parameter set values for each circuit architecture, and evaluated the objective function values of $J_2(\boldsymbol{\theta}_1)$. The resulting empirical distributions are then plotted using violin-plots [60] as shown in Fig 13C. The results in Fig 13C show a clear difference in the *distribution* of objective function values for odorant identity recovery between the multi-channel and single-channel inhibition models, providing conclusive evidence that the multi-channel Pre-LN inhibition is essential for semantic odor information processing—with both feedforward and feedback inhibition being equally effective.

## Supporting information

**S1 Appendix. S1 Table. Glossary of terms and abbreviations. S2 Table. Glossary of mathematical notation. S3 Table. Model description of the single-channel AL circuit in S2 Fig in S1 Appendix**. Note that for notational simplicity the parameter values for $\boldsymbol{\theta}$ are omitted. Indices under the *Model* column refer to the following detailed models: (1) OSN to PreLN Synapse, (2) Pre-LN BSG, (3) OSN Axon-Terminal. The Pre-LN pathway receives as input the

neurotransmitter concentration of OSN Axon-Terminal in the same channel and outputs a single Pre-LN spike train. Note that only components in the DNP-FB [I] pathway are specified as the other circuit components are the same as in Fig 10(Left) and are specified in Table 1. The single-channel Pre-LN pathway is modeled as a DNP-FB[I]. **S4 Table. Model description of the multi-channel AL circuit in S3 Fig in S1 Appendix**. Note that for notational simplicity the parameter values for θ are omitted. Indices under the *Model* column refer to the following detailed models: (1) OSN to PreLN Synapse in the *r*-th channel, (2) Pre-LN BSG, (3) OSN Axon-Terminal in the *r*-th channel. Note that only components in the Global DNP-FB pathway are specified as the other circuit components are the same as in Fig 11(Left) for each of the *R* channels. The differential equations for each model are detailed under the *Equation* column, with the corresponding parameters summarized under the *Parameters* column. Note that under the *Parameters* column, values for the fixed parameters are shown. The circuit has a total of 23 free parameters, the same as the single-channel AL circuit in **S2 Fig in S1 Appendix**. and **S3 Table in S1 Appendix**. The correspondence between the Pre-LN pathway and the Global DNP-FB model is made explicit on right side of rows (1–3). **S1 Fig. Concentration Invariance and ON/OFF Contrast Boosting of OR59b OSN and DM4 PN I/O pairs**. (A[i]) Acetone odorant concentration waveforms. (A[ii]) Acetone odorant concentration contrast computed with $\varepsilon = 1$. (A[iii]) Or59b OSN PSTHs. (A[iv]) Steady-state components of Or59b OSN PSTHs. (A[v]) Transient components of Or59b OSN PSTHs. (B[i]) Acetone odorant concentration waveforms. (B[ii]) Acetone odorant concentration contrast computed with $\varepsilon = 1$. (B[iii]) DM4 PN PSTHs. (B[iv]) Steady-state components of DM4 PN PSTHs. (B[v]) Transient components of DM4 PN PSTHs. (C[i]) Maximum cross-correlations between the odorant concentration waveforms and Or59b OSN and DM4 PN PSTHs. Maximum cross-correlation is defined as $\max_\tau \int_0^T u(t + \tau)v(t)dt$, where $u(t)$ is the odorant concentration waveform in (A[i]/B[i]) and $v(t)$ is the Or59b OSN/DM4 PN PSTH in (A[iii]/B[iii]), with $T = 17.5$ seconds (duration of the signals $u(t)$, $v(t)$) and $\tau \in [-500, 500]$ milliseconds. Each black dot in (C[i]) corresponds to a maximum cross-correlation of a single experiment in (A)/(B). (C[ii]) Same as (C[i]) but for the odorant concentration waveforms and steady-state components of Or59b OSN/DM4 PN PSTHs. (C[iii]) Same as (C[i]) but for the transient components of the odorant concentration contrast and Or59b OSN/DM4 PN PSTHs. Note that, (C[i, ii]) indicate that both the overall PSTH and the steady-state component of the DM4 PN PSTH response are less correlated with the concentration waveform than the Or59b OSN PSTH response is correlated with the concentration waveform, suggesting concentration-invariance at the PN level. (C[iii]) indicates that the transient component of DM4 PN response is more correlated with the concentration contrast than the Or59b OSN response is correlated with the concentration contrast, suggesting contrast-boosting at the PN level. **S2 Fig. Schematics of the single-channel AL circuit with Pre-LN feedback inhibition and the associated DNP circuit**. (Left) Example single-channel AL circuit with Pre-LN feedback inhibition and (Right) the associated DNP circuit. Refer to **S3 Table in S1 Appendix**. for model details. **S3 Fig. Schematics of the multi-channel AL circuit with Pre-LN global feedback inhibition and the associated DNP circuit**. (Left) Example multi-channel AL circuit with Pre-LN global feedback pathways and (Right) the associated DNP circuit. The multi-channel AL circuit has *R* channels, receiving input from OSNs expressing *R* receptor types. Shown above are channels 1 and *R*. The Pre-LN pathway receives as input the neurotransmitter concentration of OSN Axon-Terminals across channels and outputs a single Pre-LN spike train. As opposed to the cross-channel integration of the Pre-LN pathway, the Post-eLN and Post-iLN pathways are repeated for each channel and only perform computation local to the corresponding glomerulus. The same distinction of global vs. local computation is shown for the equivalent

differential DNP circuit on the right. Here the multi-channel Pre-LN pathway is modeled as a *Global* DNP-FB. Refer to **S4 Table in S1 Appendix**. for model details. **S4 Fig. Stability of Steady-State Or59b OSN and DM4 PN PSTHs**. (A) Or59b OSN and DM4 PN PSTHs are shown in grey. The steady-state response component is, respectively, shown in red. (B) Coefficient of variation of the amplitudes of the piecewise constant response components (in red) of Or59b OSN and DM4 PN PSTHs. For each PSTH in (A), the amplitude of the piecewise constant response components after each jump time is recorded (e.g., 8 such values in (A top-left)). The resulting coefficient of variation (C.V.) for a given experiment is shown by the red triangles in (B). The box-plots describe the distribution of the C.V. of Or59b OSN and DM4 PN steady-state response amplitudes. Note that since the identity of the stable attractor limit cycle is directly related to the amplitude of the steady-state responses, the C.V. of the steady-state response amplitudes serves as a proxy for the stability of the stable attractor limit cycles. We observe that the C.V. of the DM4 PN responses has much lower values than the ones of the OSN responses.
(PDF)

## Author Contributions

**Conceptualization:** Aurel A. Lazar, Tingkai Liu, Chung-Heng Yeh.

**Data curation:** Tingkai Liu.

**Formal analysis:** Aurel A. Lazar, Tingkai Liu, Chung-Heng Yeh.

**Funding acquisition:** Aurel A. Lazar.

**Investigation:** Aurel A. Lazar, Tingkai Liu, Chung-Heng Yeh.

**Methodology:** Aurel A. Lazar, Tingkai Liu, Chung-Heng Yeh.

**Project administration:** Aurel A. Lazar.

**Resources:** Aurel A. Lazar.

**Software:** Tingkai Liu, Chung-Heng Yeh.

**Supervision:** Aurel A. Lazar.

**Visualization:** Tingkai Liu.

**Writing – original draft:** Aurel A. Lazar, Tingkai Liu.

**Writing – review & editing:** Aurel A. Lazar, Tingkai Liu.

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
