## [Decision Letter · Decision Letter 0]

12 Sep 2022

Dear Dr. Lazar,

Thank you very much for submitting your manuscript "The functional logic of odor information processing in the Drosophila Antennal Lobe" for consideration at PLOS Computational Biology.

As with all papers reviewed by the journal, your manuscript was reviewed by members of the editorial board and by several independent reviewers. In light of the reviews (below this email), we would like to invite the resubmission of a significantly-revised version that takes into account the reviewers' comments.

We cannot make any decision about publication until we have seen the revised manuscript and your response to the reviewers' comments. Your revised manuscript is also likely to be sent to reviewers for further evaluation.

Sincerely,

Alexandre V. Morozov, Ph.D.

Academic Editor

PLOS Computational Biology

Wolfgang Einhäuser

Section Editor

PLOS Computational Biology

Reviewer's Responses to Questions

**Comments to the Authors:**

Reviewer #1: The review is uploaded as an attachment

Reviewer #2: My review is uploaded as an attachment

**Have the authors made all data and (if applicable) computational code underlying the findings in their manuscript fully available?**

Reviewer #1: Yes

Reviewer #2: Yes

PLOS authors have the option to publish the peer review history of their article (what does this mean?). If published, this will include your full peer review and any attached files.

Reviewer #1: No

Reviewer #2: **Yes: **Alexander Shakeel Bates
---

## [Decision Letter · Decision Letter 1]

30 Jan 2023

Dear Dr. Lazar,

Thank you very much for submitting your manuscript "The Functional Logic of Odor Information Processing in the *Drosophila* Antennal Lobe" for consideration at PLOS Computational Biology. As with all papers reviewed by the journal, your manuscript was reviewed by members of the editorial board and by several independent reviewers. The reviewers appreciated the attention to an important topic. Based on the reviews, we are likely to accept this manuscript for publication, providing that you modify the manuscript according to the review recommendations.

Sincerely,

Alexandre V. Morozov, Ph.D.

Academic Editor

PLOS Computational Biology

Wolfgang Einhäuser

Section Editor

PLOS Computational Biology

Reviewer's Responses to Questions

**Comments to the Authors:**

Reviewer #1: Review comments are in the attachment

Reviewer #2: I thank Lazar and colleagues for their work in addressing my concerns and those of reviewer 1. In particular, in improving the readability of the manuscript. Having now read comments from reviewer 1, the authors’ rebuttal and their revised manuscript I would like to re-hash one of my prior opinions rather strongly. I will not require that it needs be addressed for the paper to proceed - but I very strongly recommend to the authors that it should be, in service to the utility and impact of their work at a time for the field where linking mechanism and computation to identified neuronal hardware has never been easier and more fruitful.

I am not convinced by the authors’ claim that they do not need to consider available connectome data in this work. Reviewer 1 suggested using the connectome very broadly, and the authors replied that this was not in scope. I agree with them on that front, but a simple connectome analysis should be done to at the very least contextualise their work. I raised this point strongly in my first review, and the authors essentially ignored it. In response to my points on connectome, anatomy, and prior literature, the authors did not answer my questions but told me that they are coming at the problem from a computational logic/theory point of view, and so use abstractions inspired by the systems neuroscience view but not necessarily linked to it. I understand this perfectly well and do not take issue with any of their abstractions - yet: there is no point in describing an abstract computation for the fly antennal lobe if the fly antennal lobe does not contain the wetware to run it. Indeed, figures 3 and 4 show model architectures that can be directly linked to the neurobiological network diagram - it is a small step!

The paper is about how the fly olfactory system - not something more abstract than that - and is based on anatomical connections observable within that system. The biological constraints are important. Very important, and the easier constraints to understand come from network structure - the authors have already tried hard in the more difficult case of neurophysiology. I am not asking the authors to use connectome-derived weights or exact neuron identities in their work, or build any kind of mechanistic model. Nor am I asking for a higher level characterisation of the network in terms of feedback loops and similar, as they present in another of their bioRxiv papers that they reference (rebuttal reference 3). Rather, it is totally in keeping with the nature of the work at hand to ask whether the motifs Lazar et al. are studying actually exist and in what proportions, and to what identified cell types they pertain - none of this is dealt with in reference 3. The latter point is instrumental in providing a starting point for experimentation that might validate some of this modelling work. I am asking for a simple descriptive summary - in its simplest form, this might be some histograms or stacked bar plots showing the incidence of pre/post i/e LN-OSN connectivity. This is not about a clash of perspectives, it is about integration. To do otherwise at this time in the field is to knowingly limit the impact and usability of this work. To be clear again, I am not asking them to change anything about how their modelling is done. It is context I am after.

For example, does the connectome contain neurons that can be arranged as required for the authors’ DNPs? How frequently do you find pre-LN, post-iLN and post-eLN innervation patterns, and does this differ between glomeruli? Do eLN-OSN inputs exist at all, or are they not important? Reviewer 1 also asked why the authors “eliminate presynaptic excitatory LNs to the terminal of OSN axons”. I found the authors’ reply, essentially that they did not need to, lacking. It is a clear and singular complement to the motifs they have already chosen. Its absence is conspicuous. Are there excitatory LN inputs to OSN axons in the fly brain? The authors say they ‘we believe that, algorithmically, it is beneficial for the LN pathways to process syntactic information and semantic information independently.’ Does this pathway separation anatomically exist?

The sum of my point here is: the authors put a lot of weight into trying different network configurations and showing us and explaining the model architecture and its operations but none into telling us - given the open source data available to them via Scheffer et al. and the annotations from Schlegel et al. - whether the fly has the neuronal architectures to support it.

**Have the authors made all data and (if applicable) computational code underlying the findings in their manuscript fully available?**

Reviewer #1: None

Reviewer #2: Yes

PLOS authors have the option to publish the peer review history of their article (what does this mean?). If published, this will include your full peer review and any attached files.

Reviewer #1: No

Reviewer #2: **Yes: **Alexander Shakeel Bates

Figure Files:

Data Requirements:

Reproducibility:

References:

---

## [Editor Report · Decision Letter 2]

22 Mar 2023

Dear Dr. Lazar,

We are pleased to inform you that your manuscript 'The Functional Logic of Odor Information Processing in the *Drosophila* Antennal Lobe' has been provisionally accepted for publication in PLOS Computational Biology.

Best regards,

Alexandre V. Morozov, Ph.D.

Academic Editor

PLOS Computational Biology

Wolfgang Einhäuser

Section Editor

PLOS Computational Biology

---

## [Editor Report · Acceptance letter]

14 Apr 2023

PCOMPBIOL-D-22-00311R2 

The Functional Logic of Odor Information Processing in the *Drosophila* Antennal Lobe

Dear Dr Lazar,

I am pleased to inform you that your manuscript has been formally accepted for publication in PLOS Computational Biology. Your manuscript is now with our production department and you will be notified of the publication date in due course.

With kind regards,

Anita Estes
